# On the Tradeoff Between Robustness and Fairness

**Xinsong Ma**
School of Computer Science
Wuhan University
maxinsong1018@gmail.com

**Zekai Wang**
School of Computer Science
Wuhan University
wzekai99@gmail.com

**Weiwei Liu**[*]
School of Computer Science
Wuhan University
liuweiwei863@gmail.com

## Abstract

Interestingly, recent experimental results [2, 26] have identified a robust fairness phenomenon in adversarial training (AT), namely that a robust model well-trained by AT exhibits a remarkable disparity of standard accuracy and robust accuracy among different classes compared with natural training. However, the effect of different perturbation radii in AT on robust fairness has not been studied, and one natural question is raised: does a tradeoff exist between average robustness and robust fairness? Our extensive experimental results provide an affirmative answer to this question: with an increasing perturbation radius, stronger AT will lead to a larger class-wise disparity of robust accuracy. Theoretically, we analyze the class-wise performance of adversarially trained linear models with mixture Gaussian distribution. Our theoretical results support our observations. Moreover, our theory shows that adversarial training easily leads to more serious robust fairness issue than natural training. Motivated by theoretical results, we propose a fairly adversarial training (FAT) method to mitigate the tradeoff between average robustness and robust fairness. Experimental results validate the effectiveness of our proposed method.

## 1 Introduction

Deep neural networks (DNN) have achieved impressive performance on numerous challenging tasks ranging from computer vision [10, 16] and natural language processing [9, 11] to autonomous driving. Despite this success, it turns out that DNN is vulnerable to adversarial examples. In other words, very small and even imperceptible perturbations are sufficient to deceive DNN models, resulting in erroneous predictions [7, 8, 12, 19]. As one of the most popular methods against adversarial examples, adversarial training (AT) can effectively improve model's robustness [17, 21, 24, 29].

Recently, numerous authors [2, 26] have identified an interesting robust fairness phenomenon in AT, specifically that a robust model well-trained by AT exhibits a remarkable disparity of standard accuracy and robust accuracy among different classes compared with natural training. Usually, the performance of a naturally trained model is similar across classes. However, adversarially trained models tend to perform well in some classes and poorly in others, which will lay serious hidden dangers for some applications. For example, an autonomous driving system may achieve a satisfying average robust accuracy for recognizing objects in the road. Nevertheless, this system is robust to the classes of inanimate objects (high robust accuracy) but vulnerable to some important classes

---

[*]Corresponding author

36th Conference on Neural Information Processing Systems (NeurIPS 2022).

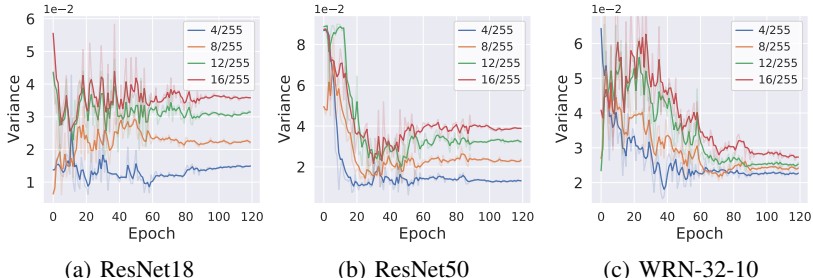

(a) ResNet18          (b) ResNet50          (c) WRN-32-10

Figure 1: The variance of class-wise robust accuracy for Madry using ResNet18, ResNet50 and WRN-32-10 on CIFAR-10. The perturbation radii for AT are chosen from $\epsilon_{train} = \{4/255, 8/255, 12/255, 16/255\}$. The adversarial testing examples are generated by FGSM with testing perturbation radius $\epsilon_{test} = 16/255$.

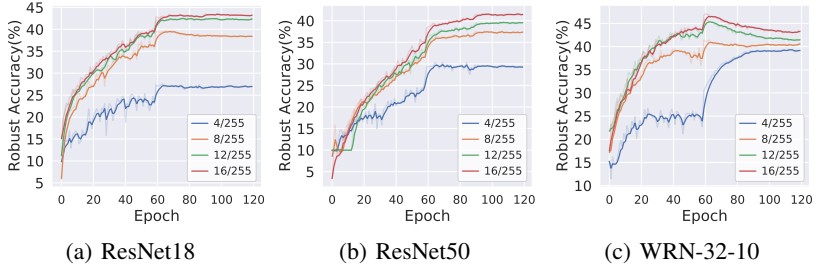

(a) ResNet18          (b) ResNet50          (c) WRN-32-10

Figure 2: The average robust accuracy for Madry using ResNet18, ResNet50 and WRN-32-10 on CIFAR-10. The perturbation radii for AT are chosen from $\epsilon_{train} = \{4/255, 8/255, 12/255, 16/255\}$. The adversarial testing examples are generated by FGSM with testing perturbation radius $\epsilon_{test} = 16/255$.

such as human (low robust accuracy), which is a "snake" in the grass for drivers and pedestrians in the road. Additionally, in many sensitive fields related to social ethics, it is crucial to ensure that the models have no discriminatory behaviors toward certain classes (groups or populations). It is therefore imperative to conduct in-depth study on the robust fairness issue. Notably, the effect of different perturbation radii in AT on robust fairness has not been studied. When further considering the robust fairness phenomenon in AT, one natural question arises:

> Is there a tradeoff between average robustness and robust fairness; specifically, as the perturbation radius increases, will stronger adversarially trained models lead to a larger class-wise disparity of robust accuracy among different classes?

This paper attempts to provide an understanding of the robust fairness phenomenon in AT and study the tradeoff between average robustness and robust fairness. Following [5, 18], we use the variance of class-wise robust accuracy to quantitatively measure model's robust fairness. Our experimental results in Figures 1 and 2 provide an affirmative answer to this question, suggesting that there is a tradeoff between average robustness and robust fairness: with the increasing perturbation radius, AT is able to improve the average robust accuracy of models over classes, but AT also leads to a larger variance of class-wise robust accuracy. Theoretically, we analyze the class-wise performance of adversarially trained linear models with mixture Gaussian distribution. Our theoretical results support our observations. Moreover, our theory shows that AT easily results in more serious fairness issue compared with natural training. Motivated by our theoretical results, we propose a variance-based regularized method to mitigate the tradeoff between average robustness and robust fairness. Our core contributions can be summarized as:

- We empirically find the relation between the variance of class-wise robust accuracy and perturbation radius in AT, namely that AT with larger perturbation radius will lead to larger variance of class-wise robust accuracy while average robust accuracy is improved, which suggests that there exists a tradeoff between robustness and fairness;

- We theoretically analyze this new phenomenon above and provide a potential explanation for it through linear model with mixture Gaussian distribution. Additionally, we theoretically prove that AT can lead to severer robust fairness issue compared with natural training;

- We propose FAT to mitigate the tradeoff between robustness and fairness, and experimental results validate the effectiveness of our proposed method.

## 2 Related Work

**Adversarial training** Madry [17] proposes adversarial training (AT) to improve the robustness of the model against adversarial examples. Some variants of AT have also been presented. For example, [21, 29] propose different regularization terms. [4, 22] add unlabeled data to the training set. [25] propose an algorithm to accelerate training process. A large number of works [6, 14, 23] have studied the tradeoff between robustness and accuracy. Among these, TRADES [29] is one of the most popular methods due to its promising experimental results. We take the two classical methods, Madry and TRADES, as our benchmarks.

**Robust fairness** Recently, [2, 20, 26] have identified the phenomenon of class-wise disparity of robustness and describe it as a robust fairness problem. [26] point out that AT can lead to larger class-wise disparity of standard accuracy compared with natural training and focus on adversarial training algorithms. Different from [2, 20, 26], this paper focuses on exploring the relation between the perturbation radius in AT and class-wise disparity of robust accuracy, and further studies whether there is a tradeoff between average robust accuracy and class-wise disparity of robust accuracy. We provide an affirmative answer to this question by comprehensive experiments and rigorous theoretical analysis.

## 3 Notation and Definition

We donote by $\mathcal{X} \subseteq \mathbb{R}^d$ the feature space and $\mathcal{Y} = \{1, 2, 3, \ldots, L\}$ the label space with unknown joint distribution $\mathcal{D}$, and $\mathcal{X}$ has marginal distribution $\mathcal{D}_x$. Let $(x, y)$ be the feature-label pair, where instance $x \in \mathcal{X}$ and label $y \in \mathcal{Y}$. We denote the transpose of vector/matrix by the superscript $'$. Denote by $f_\theta : \mathcal{X} \to \mathcal{Y}$ the classifier (or hypothesis) which is parameterized by $\theta \in \Theta$, where $\Theta$ is the parameter space and maps $x$ to a certain label $y$. For simplicity, we denote $\ell(\theta, x, y) = \ell(f_\theta(x), y)$ where $\ell(\cdot)$ is a certain loss function.

**Natural training.** Recall that for natural training in the canonical classification setting, the primary goal is to maximize the standard accuracy of models on the unseen examples from the underlying distribution. Concretely, we hope to find the classifier $f_{nat}$ with the smallest natural risk:

$$R_{nat}(f_{nat}) = \min_{\theta \in \Theta} \mathbb{E}_{(x,y) \sim \mathcal{D}} \ell(\theta, x, y).$$

**Adversarial training.** AT [17, 21, 29] aims to build models that are robust against adversarial examples by solving a robust optimization problem. Specifically, the task of AT is to find the classifier $f_{adv}$ with the smallest adversarial/robust risk:

$$R_{adv}(f_{adv}) = \min_{\theta \in \Theta} \mathbb{E}_{(x,y) \sim \mathcal{D}}[\sup_{z \in \mathcal{U}(x,\epsilon)} \ell(\theta, z, y)]$$

where $\mathcal{U}(x, \epsilon)$ represents the perturbation set. This paper focuses on the set $\mathcal{U}(x, \epsilon) = \{z \in \mathcal{X} : ||z - x||_\infty \le \epsilon\}$, where $|| \cdot ||_\infty$ is the $l_\infty$-norm. Such type of set is also called $l_\infty$-attack in AT.

## 4 Empirical Exploration

This section explores the relations between the perturbation radius in AT and the variance of class-wise robust accuracy, and studies whether there is a tradeoff between average robust accuracy and class-wise disparity of robust accuracy through a series of experiments. We use Madry [17] with various perturbation radii $\epsilon_{train} = \{4/255, 8/255, 12/255, 16/255\}$ to train ResNet18, ResNet50 [13] and Wide ResNet-32-10 (WRN-32-10) [28] on the CIFAR-10 and CIFAR-100 datasets [15]. The evaluation metrics are the average robust accuracy and the variance of class-wise robust accuracy. The adversarial testing examples are generated by FGSM [12], PGD-20 [17] and C&W [3] with different testing perturbation radii $\epsilon_{test} = \{16/255, 20/255\}$, respectively. The experimental settings(including optimizer, epoch, learning rate and so on) are same as Section 7.1 Here, we present only the experimental results of Madry on CIFAR-10.

Table 1: The average robust accuracy (%) and variance of class-wise robust accuracy ($10^{-2}$) for Madry using ResNet18, ResNet50 and WRN-32-10 on CIFAR-10. The perturbation radii for AT are chosen from $\epsilon_{train} = \{4/255, 8/255, 12/255, 16/255\}$. The adversarial testing examples are generated by FGSM, PGD-20 and C&W with different testing perturbation radii $\epsilon_{test} = \{16/255, 20/255\}$. We use var and rob.acc to denote the variance of class-wise robust accuracy, and average robust accuracy, respectively.

| attack algorithm | | FGSM | | | | PGD-20 | | | | C&W | | | |
| --- | --- | --- | --- | --- | --- | --- | --- | --- | --- | --- | --- | --- | --- |
| $\epsilon_{test}$ | | 16/255 | | 20/255 | | 16/255 | | 20/255 | | 16/255 | | 20/255 | |
| model | $\epsilon_{train}$ | var | rob.acc | var | rob.acc | var | rob.acc | var | rob.acc | var | rob.acc | var | rob.acc |
| ResNet18 | 4/255 | 1.90 | 29.04 | 1.31 | 21.71 | 0.10 | 3.54 | 0.02 | 1.12 | 0.11 | 3.61 | 0.02 | 1.05 |
| | 8/255 | 2.34 | 39.00 | 1.99 | 32.13 | 0.50 | 11.23 | 0.16 | 4.96 | 0.61 | 12.14 | 0.19 | 5.35 |
| | 12/255 | 3.12 | 41.99 | 2.53 | 35.71 | 0.98 | 19.43 | 0.35 | 10.03 | 1.09 | 19.39 | 0.42 | 10.53 |
| | 16/255 | 3.52 | 42.94 | 2.99 | 37.34 | 1.28 | 22.50 | 0.48 | 12.86 | 1.38 | 22.03 | 0.60 | 12.65 |
| ResNet50 | 4/255 | 1.24 | 28.89 | 0.84 | 21.14 | 0.07 | 3.34 | 0.01 | 1.13 | 0.07 | 3.47 | 0.01 | 1.08 |
| | 8/255 | 2.36 | 37.74 | 1.81 | 30.68 | 0.56 | 13.13 | 0.22 | 6.10 | 0.68 | 14.06 | 0.29 | 6.90 |
| | 12/255 | 3.15 | 39.28 | 2.78 | 33.95 | 1.07 | 20.88 | 0.38 | 11.59 | 1.16 | 20.19 | 0.48 | 11.70 |
| | 16/255 | 3.89 | 41.26 | 3.20 | 35.95 | 1.54 | 23.92 | 0.66 | 14.44 | 1.61 | 22.56 | 0.75 | 14.11 |
| WRN-32-10 | 4/255 | 2.30 | 39.21 | 0.16 | 5.15 | 0.16 | 5.15 | 0.02 | 1.79 | 0.09 | 3.97 | 0.01 | 1.10 |
| | 8/255 | 2.46 | 40.68 | 0.93 | 13.62 | 0.93 | 13.62 | 0.43 | 7.91 | 0.74 | 12.76 | 0.25 | 6.58 |
| | 12/255 | 2.58 | 41.51 | 0.97 | 15.95 | 0.96 | 15.63 | 0.48 | 9.09 | 0.96 | 15.69 | 0.46 | 8.87 |
| | 16/255 | 2.73 | 43.36 | 1.04 | 17.05 | 1.04 | 17.05 | 0.65 | 9.87 | 1.00 | 17.09 | 0.53 | 9.76 |

The results in Figures 1 and 2 show that with increasing $\epsilon_{train}$, the average robust accuracy for Madry using three neural network models on CIFAR-10 monotonously increases; moreover, their variance of class-wise robust accuracy monotonously increases as well, which leads to severer robust fairness issues. For example, when $\epsilon_{train} = 4/255$, the average robust accuracy and the variance of class-wise robust accuracy are $29.04\%$ and $0.019$ respectively for Madry using ResNet18 on CIFAR-10 under FGSM attack. When $\epsilon_{train} = 16/255$, Madry using ResNet18 achieves $42.94\%$ average robust accuracy, while the variance of class-wise robust accuracy of ResNet18 increases to $0.035$ on CIFAR-10. Additionally, the detailed testing results in Table 1 reveal that this trend consistently exists under different networks, attack algorithms and testing perturbation radii.

The results of Madry on CIFAR-100 are presented in Appendix A. We also use TRADES [29] with various perturbation radii $\epsilon_{train}$ to train ResNet18, ResNet50 and WRN-32-10 on CIFAR-10 and CIFAR-100. These results are presented in Appendix A. From these results, we can observe the same trend above. These observations suggest that the phenomenon in Figure 1, Figure 2 and Table 1 can be generalized to more AT algorithms, model architectures, datasets and adversarial attacks. From the above empirical analysis, we find that higher average robust accuracy is accompanied by a larger variance of class-wise robust accuracy, suggesting that there exists a tradeoff between average robustness and robust fairness.

Table 2: The robust accuracy (%), average robust accuracy (%) and variance of class-wise robust accuracy ($10^{-2}$) of some classes for Madry using ResNet50 on CIFAR-10. The perturbation radii for AT are chosen from $\epsilon_{train} = \{4/255, 8/255, 12/255, 16/255\}$. The adversarial testing examples are generated by FGSM with testing perturbation radius $\epsilon_{test} = 16/255$. We use var and rob.acc to denote the variance of class-wise robust accuracy, and average robust accuracy, respectively. Moreover, diff means the difference between testing results where $\epsilon_{train}$ is $16/255$ and $4/255$, respectively.

| $\epsilon_{train}$ | cat | deer | horse | ship | var | rob.acc. |
| --- | --- | --- | --- | --- | --- | --- |
| 4/255 | 12.0 | 17.5 | 39.9 | 34.2 | 1.24 | 28.89 |
| 8/255 | 10.9 | 16.7 | 49.6 | 53.5 | 2.36 | 37.74 |
| 12/255 | 8.9 | 13.7 | 50.4 | 60.4 | 3.15 | 39.42 |
| 16/255 | 7.8 | 13.7 | 51.4 | 62.4 | 3.89 | 41.26 |
| diff | -4.2 | -3.8 | 11.5 | 28.2 | 2.65 | 12.37 |

Why does AT results in this phenomenon? Table 2 presents the robust accuracy of some classes, average robust accuracy and variance of class-wise robust accuracy for Madry using ResNet50 on CIFAR-10. The results in Table 2 may reveal the underlying rationale for our findings. From Table 2, we can observe that model's robust accuracy on classes "cat" and "deer" is significantly lower than that on classes "horse" and "ship" for various values of $\epsilon_{train}$. Additionally, with the increasing of $\epsilon_{train}$, the robust accuracy of the model on the classes "cat" and "deer" monotonously decreases; while the robust accuracy of the model on classes "horse" and "ship" monotonously increases. This observation indicates that, to further improve the robustness of models, AT prefers to "sacrifice" model's robust accuracy on relatively vulnerable classes (like "cat" and "deer") and pay more atten-

tion to learning the distribution of examples of more robust classes (like 'horse' and "ship"). This preference of AT increases the variance of class-wise robust accuracy and leads to a tradeoff between average robustness and robust fairness, although the average robustness of the model can be improved. The following section presents a theoretical analysis of this phenomenon.

# 5   Theoretical Analysis

Following [4, 22, 26], we study a concrete mixture Gaussian distribution. Inspired by the results in Table 2, our motivation is to design two classes, "+1" and "−1" and then we analyze the performance of natural training and AT on each class. We begin with the following definition.

**Definition 5.1.** (Mixture Gaussian Distribution). Let $\mu_+, \mu_- > 0$ be the per-class mean parameter and $\sigma_+, \sigma_- > 0$ be variance parameter of two classes. The $(\mu_+, \mu_-, \sigma_+, \sigma_-)$-Gaussian mixture distribution $\mathcal{D}^*$ can be then defined by the following distribution over $(x, y) \in \mathbb{R}^d \times \{\pm 1\}$:

$$y = \begin{cases} +1, & p = \alpha \\ -1, & p = 1 - \alpha, \end{cases} \qquad x \sim \begin{cases} \mathcal{N}\left(\boldsymbol{\mu}_+, \sigma_+^2 I\right) & \text{if } y = +1 \\ \mathcal{N}\left(-\boldsymbol{\mu}_-, \sigma_-^2 I\right) & \text{if } y = -1 \end{cases} \tag{1}$$

where $\alpha$ is the prior probability of class "+1" and $\boldsymbol{\mu}_+ = \mu_+ \mathbf{1}, \boldsymbol{\mu}_- = \mu_- \mathbf{1}, \mathbf{1} = \overbrace{(1, \ldots, 1)}^{\dim d}{}', I$ is a $d$- dimension identity matrix.

In this section, we focus on studying the performance of the linear model on the distribution $\mathcal{D}^*$ as follows:

$$f(x) = \text{sign}\left(\langle w, x \rangle + b\right) \tag{2}$$

where the parameters $w \in \mathbb{R}^d, b \in \mathbb{R}$, and $\text{sign}(\text{t})$ evaluates to 1 if scalar $t \geq 0$ and to -1 otherwise.

## 5.1   Naturally Trained Linear Model

In this subsection, we analyze the performance of a naturally trained linear model on distribution $\mathcal{D}^*$. For simplicity, we denote

$$R_{nat}^{+1}(f) = \mathbb{E}_{(x,y)\sim\mathcal{D}^*}(\mathbb{1}(f(x) = -1)|y = +1)$$

$$R_{nat}^{-1}(f) = \mathbb{E}_{(x,y)\sim\mathcal{D}^*}(\mathbb{1}(f(x) = +1)|y = -1)$$

where $\mathbb{1}(\cdot)$ is the indicator function that takes 1 when the statement in the braces is true and 0 otherwise.

**Theorem 5.2.** *For the naturally trained linear classifier $f_{nat}$ that minimizes the natural risk:*

$$f_{nat}(x) = \arg\min_f \mathbb{E}_{(x,y)\sim\mathcal{D}^*}(\mathbb{1}(f(x) \neq y)), \tag{3}$$

*if $\sigma_+ \neq \sigma_-$, the class-wise natural risk can be expressed as follows:*

$$R_{nat}^{+1}(f_{nat}) = \Phi\left(\frac{-\eta^* - d\mu_+}{\sqrt{d}\sigma_+}\right) \quad R_{nat}^{-1}(f_{nat}) = \Phi\left(\frac{\eta^* - d\mu_-}{\sqrt{d}\sigma_-}\right) \tag{4}$$

*where $\Phi(\cdot)$ is the cumulative distribution function (c.d.f.) of standard Gaussian distribution $\mathcal{N}(0, 1)$ and*

$$\eta^* = \frac{A + \sigma_+\sigma_-(\mu_+ + \mu_-)\sqrt{1 + 2K\frac{\sigma_-^2 - \sigma_+^2}{(\mu_+ + \mu_-)^2}}}{\sigma_-^2 - \sigma_+^2}$$

*where $A = -d(\mu_+\sigma_-^2 + \mu_-\sigma_+^2)$ and $K$ is a positive constant. If $\sigma_+ = \sigma_- = \sigma$, the class-wise natural risk can be expressed as follows:*

$$R_{nat}^{+1}(f_{nat}) = \Phi\left(\frac{-d^2(\mu_+ + \mu_-)^2 - 2K\sigma^2}{2d^{3/2}\sigma(\mu_+ + \mu_-)}\right) \tag{5}$$

$$R_{nat}^{-1}(f_{nat}) = \Phi\left(\frac{-d^2(\mu_+ + \mu_-)^2 + 2K\sigma^2}{2d^{3/2}\sigma(\mu_+ + \mu_-)}\right). \tag{6}$$

A proof of Theorem 5.2 can be found in Appendix B.1. Because $\Phi(\cdot)$ is a monotonously increasing function, Theorem 5.2 shows that if the variances of two classes are same, the naturally trained model performs better on class "+1" than class "−1"; namely:

$$R_{nat}^{+1}(f_{nat}) < R_{nat}^{-1}(f_{nat}).$$

## 5.2 Adversarially Trained Linear Model

This subsection studies the performance of the adversarially trained linear model on distribution $\mathcal{D}^*$. Without loss of generality, we consider the $l_\infty$-attack in adversarial training and testing. In this case, the perturbation set can be expressed as $\mathcal{U}(x, \epsilon) = \{z : ||z - x||_\infty \le \epsilon\}$. The radius $\epsilon$ of the perturbation set can be used to adjust the level of attack. Obviously, larger $\epsilon$ means that adversary can generate stronger adversarial examples which can easily trick the trained model into making incorrect predictions.

We use $\epsilon_{train}$ and $\epsilon_{test}$ to represent the radius of the perturbation set in the training and testing phases, respectively, and we further assume $\epsilon_{train} \le \epsilon_{test}$. For simplicity, we denote

$$R_{adv}^{+1}(f) = \mathbb{E}_{(x,y)\sim\mathcal{D}^*}[\sup_{z\in\mathcal{U}(x,\epsilon)} \mathbb{1}(f(z) = -1)|y = +1]$$

$$R_{adv}^{-1}(f) = \mathbb{E}_{(x,y)\sim\mathcal{D}^*}[\sup_{z\in\mathcal{U}(x,\epsilon)} \mathbb{1}(f(z) = +1)|y = -1].$$

**Theorem 5.3.** *For the adversarial trained linear classifier $f_{adv}$ that minimizes the adversarial risk:*

$$f_{adv}(x) = \arg\min_{f}\mathbb{E}_{(x,y)\sim\mathcal{D}^*}[\sup_{z\in\mathcal{U}(x,\epsilon)} (\mathbb{1}(f(z) \ne y)], \tag{7}$$

*if $\sigma_+ \ne \sigma_-$, the class-wise adversarial risk can be expressed as follows:*

$$R_{adv}^{+1}(f_{adv}) = \Phi\left(\frac{-\gamma^* - d(\mu_+ - \epsilon_{test})}{\sqrt{d}\sigma_+}\right) \quad R_{adv}^{-1}(f_{adv}) = \Phi\left(\frac{\gamma^* - d(\mu_- - \epsilon_{test})}{\sqrt{d}\sigma_-}\right) \tag{8}$$

*where $\Phi(\cdot)$ is the c.d.f. of standard Gaussian distribution $\mathcal{N}(0, 1)$ and*

$$\gamma^* = \frac{B + \sigma_+\sigma_-(\mu_+ + \mu_- - 2\epsilon_{train})\sqrt{1 + \frac{2K(\sigma_-^2 - \sigma_+^2)}{(\mu_+ + \mu_- - 2\epsilon_{train})^2}}}{\sigma_-^2 - \sigma_+^2}$$

*where $B = -d(\mu_+\sigma_-^2 + \mu_-\sigma_+^2 - \epsilon_{train}(\sigma_+^2 + \sigma_-^2))$ and $K$ is a positive constant. If $\sigma_+ = \sigma_- = \sigma$, the class-wise adversarial risk can be expressed as follows:*

$$R_{adv}^{+1}(f_{adv}) = \Phi\left(\frac{-C - 2K\sigma^2}{2d^{3/2}\sigma(\mu_+ + \mu_- - 2\epsilon_{train})}\right)$$

$$R_{adv}^{-1}(f_{adv}) = \Phi\left(\frac{-C + 2K\sigma^2}{2d^{3/2}\sigma(\mu_+ + \mu_- - 2\epsilon_{train})}\right)$$

*where $C = d^2(\mu_+ + \mu_- - 2\epsilon_{train})(\mu_+ + \mu_- - 2\epsilon_{test})$.*

The proof of Theorem 5.3 can be seen in Appendix B.2. Theorem 5.3 demonstrates that if the variances of two classes are the same, we have

$$R_{adv}^{+1}(f_{adv}) < R_{adv}^{-1}(f_{adv}),$$

which indicates that the examples in class "$-1$" are more vulnerable than those in class "$+1$". In the following, we will present our core theoretical results based on Theorems 5.2 and 5.3.

## 5.3 Tradeoff between Average Robustness and Robust Fairness

Following [5, 18], we use *variance of class-wise robust accuracy* (VCRA) to quantitatively measure the model fairness. The definition of VCRA is presented below.

**Definition 5.4.** (VCRA). Given a classifier $f : \mathcal{X} \to \mathcal{Y}$ where $\mathcal{Y} = \{1, 2, 3, \ldots, L\}$, the variance of class-wise robust accuracy of $f$ is defined as

$$VCRA(f) = \frac{1}{L}\sum_{i=1}^{L}(p_{adv}(i) - \bar{p}_{adv})^2 \tag{9}$$

where $p_{adv}(i) = \mathbb{P}\{\forall z \in \mathcal{U}(x,\epsilon), f(z) = i|y = i\}$ and $\bar{p}_{adv} = \frac{1}{L}\sum_{i=1}^{L} p_{adv}(i)$. Additionally, we have

$$p_{adv}(i) = 1 - \mathbb{P}\{\exists z \in \mathcal{U}(x,\epsilon), f(z) \neq i|y = i\} \tag{10}$$
$$= 1 - \mathbb{E}_{(x,y)\sim\mathcal{D}}[\sup_{z\in\mathcal{U}(x,\epsilon)} \mathbb{1}(f(z) \neq i)|y = i].$$

When calculatinng the *variance of class-wise accuracy* of $f$: $VCA(f)$, we just need to replace $p_{adv}(i)$ with $p(i)$ where $p(i) = \mathbb{P}(f(x) = i|y = i)$. To obtain more concise analytical results, we focus on the situation in which $\sigma_+ \leq \sigma_-$. However, our results can be generalized to other settings under some mild conditions.

**Theorem 5.5.** *Given an adversarially trained linear model $f_{adv}$ in Equation* (7)*, the variance of class-wise robust accuracy $VCRA(f_{adv})$ is increasing with respect to $\epsilon_{train}$ if $\sigma_- \geq \sigma_+$.*

The proof of Theorem 5.5 can be found in Appendix B.3. A large radius $\epsilon_{test}$ of the perturbation set $\mathcal{U}(x,\epsilon_{test})$ in the testing phase enables adversary to generate strong adversarial examples that can easily trick the trained model into making wrong predictions. To defend against these adversarial examples, we usually increase the radius $\epsilon_{train}$ of $\mathcal{U}(x,\epsilon_{train})(\epsilon_{train} \leq \epsilon_{test})$ to generate strong adversarial examples in the training phase, which has been proved to effectively improve the average robust accuracy of models. However, Theorem 5.5 shows that the variance of the class-wise robust accuracy of the adversarially trained model $f_{adv}$ increases monotonically with respect to the radius $\epsilon_{train}$. In other words, a larger radius $\epsilon_{train}$ will lead to a larger variance of class-wise robust accuracy of the adversarially trained model even though the average robust accuracy of the model might be improved. Therefore, there exists a tradeoff between average robust accuracy and robust fairness. This provides an confirmative answer to our question in Section 1 and explains our experimental findings in Section 4. Notably, when perturbation radius $\epsilon_{test} = 0$, Theorem 5.5 suggests that AT($\epsilon_{train} > 0$) results in larger class-wise disparity of standard accuracy compared with natural training($\epsilon_{train} = 0$). Hence, the results in [26] is a special case of Theorem 5.5.

Why does AT lead to this phenomenon? For simplicity, we just consider the situation in which $\sigma_- = \sigma_+$. In this setting, $R_{adv}^{+1}(f_{adv})$ is monotonically decreasing, but $R_{adv}^{-1}(f_{adv})$ is monotonically increasing with respect to $\epsilon_{train}$. Therefore, improving model's average robust accuracy by increasing $\epsilon_{train}$ will result in a smaller $R_{adv}^{+1}(f_{adv})$ and larger $R_{adv}^{-1}(f_{adv})$, meaning larger class-wise disparity of robust accuracy. Moreover, the examples in class "$-1$" are more vulnerable than those in class "$+1$". These analysis suggests that, to further improve average robust accuracy, AT will "sacrifice" model's robust accuracy on relatively vulnerable class "$-1$" and pay more attention to learning the distribution of examples in more robust class "$+1$", which explains our observations in Table 2. This preference of AT will lead to a tradeoff between average robust accuracy and robust fairness.

Previous works [2, 26] empirically find that a robust model well-trained by AT exhibits a remarkable disparity of standard accuracy and robust accuracy among different classes compared with natural training. We will theoretically prove this phenomeno. First, we give following theoretical result.

**Theorem 5.6.** *Given an adversarially trained linear model $f_{adv}$ in Equation* (7)*, and suppose $\sigma_+ = \sigma_-$. The variance of class-wise robust accuracy of $f_{adv}$ can be expressed as follows:*

$$VCRA(f_{adv}) = \frac{1}{\pi d^3}\exp(-\xi^2)\left(\frac{K\sigma}{\mu_+ + \mu_- - 2\epsilon_{train}}\right)^2$$

*where $\xi$ is a constant and $K$ is a positive constant.*

The proof of Theorem 5.6 can be found in Appendix B.3.

**Theorem 5.7.** *Given an adversarially trained linear model $f_{adv}$ in Equation* (7) *and a naturally trained linear model $f_{nat}$ in Equation* (3)*, and suppose $\sigma_+ = \sigma_-$. If $\epsilon_{train} < \frac{\mu_+ + \mu_-}{e}$ and $\epsilon_{test} < \frac{\mu_+ + \mu_-}{2} - \frac{eK^3\sigma^3}{d^4(\mu_+ + \mu_-)}$, we have*

$$\frac{VCRA(f_{adv})}{VCRA(f_{nat})} = e^{(\zeta_2^2 - \zeta_1^2)}\left(\frac{\mu_+ + \mu_-}{\mu_+ + \mu_- - 2\epsilon_{train}}\right)^2 \geq 1$$

*where $K$ is a positive constant and $\zeta_1$, $\zeta_2$ are two constants.*

The proof of Theorem 5.7 can be found in Appendix B.3. From Theorem 5.7, we can find $VCRA(f_{nat}) < VCRA(f_{adv})$. Similar to Theorem 5.7, we also demonstrate $VCA(f_{nat}) \leq VCRA(f_{adv})$ in Theorem B.6 of Appendix B.3. The results of Theorems 5.7 and B.6 suggest that AT easily leads to severer robust fairness issue than natural training.

## 6  Fairly Adversarial Training

In this section, we will propose a novel fairly adversarial training (FAT) to mitigate the tradeoff between average robustness and robust fairness. The analytical results in Section 5 and Equation (10) show that we can control the variance of class-wise robust accuracy by controlling the variance of class-wise adversarial risk. Theorem 6.1 provide a theoretical guarantee for using the variance of class-wise adversarial risk to trade off average robustness against robust fairness.

Denote $\tilde{\ell}(\theta, x, y) = \sup_{z \in \mathcal{U}(x, \epsilon)} \ell(\theta, z, y)$ and the corresponding empirical adversarial risk $\hat{R}_{adv}(f) = \frac{1}{n} \sum_{i=1}^{n} \tilde{\ell}(\theta, x_i, y_i)$. In addition, we denote the variance of class-wise adversarial risk (VCAR) as

$$\text{VCAR}(f) = \frac{1}{L} \sum_{i=1}^{L} (R_{adv}(f, i) - \bar{R}_{adv}(f))^2$$

where $R_{adv}(f, i) = \mathbb{E}(\tilde{\ell}(\theta, x, y = i))$ and $\bar{R}_{adv}(f) = \frac{1}{L} \sum_{i=1}^{L} R_{adv}(f, i)$.

**Theorem 6.1.** *Let* $(x_1, y_1), (x_2, y_2), \ldots, (x_n, y_n)$ *be drawn independent and identically distributed (i.i.d.) from the unknown distribution* $\mathcal{D}$. *Under appropriate conditions on the loss* $\ell(\cdot)$, *parameter space* $\Theta$, *with probability of at least* $1 - \delta$, *the following holds for all* $\theta \in \Theta$:

$$R_{adv}(f) \leq \hat{R}_{adv}(f) + \sqrt{\frac{\text{VCAR}(f)\frac{1}{\delta}}{n}} + \frac{C}{n}$$

The proof of Theorem 6.1 can be found in Appendix B.4. Theorem 6.1 shows that the variance of class-wise adversarial risk can control the robust risk of classifiers and the class-wise disparity of robustness. Inspired by Theorem 6.1, we propose to minimize the following object:

$$\min_f [\underbrace{R_{adv}(f)}_{\text{for robustness}} + \underbrace{\lambda \text{VCAR}(f)}_{\text{for disparity}}] \tag{11}$$

where the parameter $\lambda$ in Equation (11) captures the tradeoff between average robustness and robust fairness.

We use $\hat{R}_{adv}(f, i)$ to estimate $R_{adv}(f, i)$ where $\hat{R}_{adv}(f, i) = \frac{1}{n_i} \sum_{j=1}^{n_i} \tilde{\ell}(\theta, x_j, i)$ and $n_i$ is the number of examples that belong to class $i$. The estimator of $\text{VCAR}(f)$ can be expressed as follows:

$$\widehat{\text{VCAR}}(f) = \frac{1}{L} \sum_{i=1}^{L} (\hat{R}_{adv}(f, i) - \bar{\hat{R}}_{adv}(f))^2$$

Usually, $\widehat{\text{VCAR}}(f)$ is a biased but consistent estimator of $VCAR(f)$. FAT can thus be formulated as follows:

$$\min_f [\hat{R}_{adv}(f) + \lambda \widehat{\text{VCAR}}(f)] \tag{12}$$

## 7  Experiments

In this section, we present the experimental results to validate the effectiveness of FAT for mitigating the tradeoff between average robustness and robust fairness.

### 7.1  Experimental Settings

We conduct our experiments on the benchmark datasets CIFAR-10 and CIFAR-100 [15]. Based on ResNet18 [13], we compare our proposed FAT with Madry [17] on two datasets with different

Table 3: The variance of class-wise robust accuracy ($10^{-2}$) for Madry and FAT using ResNet18 on CIFAR-10. The perturbation radii are chosen from $\epsilon_{train} = \{4/255, 8/255, 12/255, 16/255\}$. The adversarial testing examples are generated by FGSM, PGD-20 and C&W with different testing perturbation radii $\epsilon_{test} = \{16/255, 20/255\}$.

| attack algorithm | FGSM | | | | PGD-20 | | | | C&W | | | |
|---|---|---|---|---|---|---|---|---|---|---|---|---|
| $\epsilon_{test}$ | 16/255 | | 20/255 | | 16/255 | | 20/255 | | 16/255 | | 20/255 | |
| $\epsilon_{train}$ | PGD | FAT | PGD | FAT | PGD | FAT | PGD | FAT | PGD | FAC | PDG | FAT |
| 4/255 | 1.90 | 1.76 | 1.31 | 1.28 | 0.10 | 0.07 | 0.02 | 0.01 | 0.11 | 0.08 | 0.02 | 0.01 |
| 8/255 | 2.34 | 2.18 | 1.99 | 1.69 | 0.50 | 0.46 | 0.16 | 0.12 | 0.61 | 0.52 | 0.19 | 0.14 |
| 12/255 | 3.12 | 2.99 | 2.53 | 2.47 | 0.98 | 0.84 | 0.35 | 0.35 | 1.09 | 0.90 | 0.42 | 0.38 |
| 16/255 | 3.52 | 3.32 | 2.99 | 2.90 | 1.28 | 1.06 | 0.48 | 0.40 | 1.38 | 1.22 | 0.60 | 0.51 |

Table 4: The average robust accuracy ($10^{-2}$) for Madry and FAT using ResNet18 on CIFAR-10. The perturbation radii are chosen from $\epsilon_{train} = \{4/255, 8/255, 12/255, 16/255\}$. The adversarial testing examples are generated by FGSM, PGD-20 and C&W with different testing perturbation radii $\epsilon_{test} = \{16/255, 20/255\}$. We use **bold** to denote higher robust accuracy.

| attack algorithm | FGSM | | | | PGD-20 | | | | C&W | | | |
|---|---|---|---|---|---|---|---|---|---|---|---|---|
| $\epsilon_{test}$ | 16/255 | | 20/255 | | 16/255 | | 20/255 | | 16/255 | | 20/255 | |
| $\epsilon_{train}$ | PGD | FAT | PGD | FAT | PGD | FAT | PGD | FAT | PGD | FAC | PDG | FAT |
| 4/255 | 29.04 | **29.33** | 21.74 | **22.65** | 3.54 | 3.11 | 1.12 | **1.31** | 3.61 | 3.23 | 1.05 | 0.99 |
| 8/255 | 39.00 | 38.57 | 32.13 | **32.26** | 11.23 | **11.29** | 4.96 | **5.26** | 12.14 | **12.32** | 5.35 | **5.48** |
| 12/255 | 41.99 | **42.77** | 35.71 | **36.71** | 19.43 | **19.74** | 10.03 | **10.97** | 19.39 | **19.88** | 10.53 | **11.48** |
| 16/255 | 42.97 | **43.40** | 37.34 | **37.98** | 22.50 | **22.79** | 12.86 | **13.11** | 22.03 | **22.21** | 12.65 | **12.99** |

perturbation radii $\epsilon_{train} \in \{4/255, 8/255, 12/255, 16/255\}$. The maximum PGD step and step size are set to 20 and $\epsilon_{train}/10$, respectively. For optimization, we use SGD with 0.9 momentum for 120 epochs. The initial learning rate is set to 0.1 and is divided by 10 at epoch 60 and epoch 80, respectively. The adversarial testing examples are generated by FGSM [12], PGD-20 [17] and C&W [3] with different testing perturbation radii $\epsilon_{test} = \{16/255, 20/255\}$, respectively. Here, we present only the experimental results on CIFAR-10 in Tables 3 and 4. The experiment results of FAT and Madry on CIFAR-100 can be found in Appendix C.1. Besides, the sensitivity analysis of the regularization hyperparameter $\lambda$ in FAT is presented in Appendix C.2. Our code can be found on GitHub at `https://github.com/wzekai99/FAT`.

## 7.2 Experimental Results

Under the attack of FGSM, PGD-20 and C&W with different $\epsilon_{test}$, Table 3 presents the variance of class-wise robust accuracy for FAT and Madry using ResNet18 on CIFAR-10. From table 3, we can observe that FAT significantly reduces the variance of class-wise robust accuracy of Madry for various perturbation radii $\epsilon_{train}$. The results from Tables 3 and 4 further show that with increasing $\epsilon_{train}$, the average robust accuracy and the variance of class-wise robust accuracy of FAT increase, which is consistent with our theory in Section 5. However, compared with Madry, FAT consistently obtains smaller variance of class-wise robust accuracy while achieving comparable average robust accuracy on different $\epsilon_{train}$ and attack algorithms. For example, with the increasing of $\epsilon_{train}$ from $4/255$ to $16/255$, FAT reduces the variance of class-wise robust accuracy of Madry by around $16\%$ under the C&W attack with $\epsilon_{test} = 16/255$. These experimental results suggest that FAT can effectively suppress the growth of variance of class-wise robust accuracy while achieving an competitive average robust accuracy compared with Madry with the increasing of $\epsilon_{train}$. The results of FAT and TRADES in Appendix C.1 also present conclusions consistent with those drawn here. Therefore, FAT can alleviate the robust fairness problem and mitigate the tradeoff between average robustness and robust fairness.

# 8 Conclusion

In this work, we empirically find that there is a tradeoff between average robustness and robust fairness: higher average robust accuracy of models is accompanied by severer robust fairness issues with the increasing of training perturbation radius. Our theoretical results explain our observations. We then theoretically demonstrate that AT indeed leads to severer robust fairness issue compared with natural training. Moreover, a novel method is proposed to mitigate the tradeoff between average robustness and robust fairness. Our experimental results verify the superiority of our proposals.

## Acknowledgements

This work is supported by the National Natural Science Foundation of China under Grant 61976161.

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
