# A More Experimental Results of Empirical Exploration

In this section, we present more experimental results of our Empirical Exploration 4. From the results in Tables 5, 6 and 7, we can observe a similar phenomenon presented in Section 4, namely that higher average robust accuracy of models is accompanied by larger variance of class-wise robust accuracy(a severer robust fairness issue) with the increasing of perturbation radius $\epsilon_{train}$. In addition, this trend can be consistently observed under different neural networks, attack algorithms and datasets. These observations suggest the existence of a tradeoff between average robustness and robust fairness.

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

*Proof.* For any classifier $f(x)$ in Equation (2), we first calculate its natural risk.

$$R_{nat}(f) = \mathbb{E}_{(x,y)\sim\mathcal{D}^*}(\mathbb{1}(f(x) \neq y)) \tag{17}$$

$$= \mathbb{P}_{(x,y)\sim\mathcal{D}^*}(f(x) \neq y) \tag{18}$$

$$= \mathbb{P}(y = +1) \cdot \mathbb{P}(f(x) = -1|y = +1) + \mathbb{P}(y = -1) \cdot \mathbb{P}(f(x) = +1|y = -1) \tag{19}$$

$$= \alpha \cdot R_{nat}^{+1}(f) + (1 - \alpha) \cdot R_{nat}^{-1}(f) \tag{20}$$

where $\alpha = \mathbb{P}(y = +1)$.

Denote $x = (x_1, x_2, \ldots, x_d)'$ and $w = (w_1, w_2, \ldots, w_d)'$, we explicitly calculate $R_{nat}(f, +1)$

$$R_{nat}^{+1}(f) = \mathbb{P}(f(x) = -1|y = +1) \tag{21}$$

$$= \mathbb{P}(\langle w, x\rangle + b < 0|y = +1) = \mathbb{P}(\sum_{i=1}^{d} w_i x_i + b < 0) \tag{22}$$

where $x_1, x_2, \ldots, x_d$ are i.i.d. random variables from Gaussian distribution $\mathcal{N}(\mu_+, \sigma_+^2)$ according to the definition of $\mathcal{D}^*$ in Equation (1).

Like $R_{nat}^{+1}(f)$, we have

$$R_{nat}^{-1}(f) = \mathbb{P}(\sum_{i=1}^{d} w_i x_i + b > 0)$$

where $x_1, x_2, \ldots, x_d$ are i.i.d. from $\mathcal{N}(-\mu_-, \sigma_-^2)$. Denote $f_{nat}(x) = \langle w^*, x\rangle + b^*$. According to the method of [26], we can proof $w_1^* = w_2^* = \cdots = w_d^*$ by contradiction. Based on the properties of Gaussian distribution,

$R_{nat}^{+1}(f_{nat})$ and $R_{nat}^{-1}(f_{nat})$ can be expressed as follows:

$$R_{nat}^{+1}(f_{nat}) = \mathbb{P}(\sum_{i=1}^{d} w_i^* x_i + b < 0) = \mathbb{P}(w_1^* \sum_{i=1}^{d} x_i + b < 0)$$

$$= \mathbb{P}\left( \frac{w_1^* \sum_{i=1}^{d}(x_i - \mu_+)}{\sqrt{w_1^{*2} \sum_{i=1}^{d} \sigma_+^2}} < \frac{-b - dw_1^* \mu_+}{\sqrt{w_1^{*2} \sum_{i=1}^{d} \sigma_+^2}} \right)$$

$$= \Phi\left( -\frac{b + dw_1^* \mu_+}{\sqrt{d} w_1^* \sigma_+} \right)$$

$$R_{nat}^{-1}(f_{nat}) = \mathbb{P}(\sum_{i=1}^{d} w_i^* x_i + b > 0) = \mathbb{P}(w_1^* \sum_{i=1}^{d} x_i + b > 0)$$

$$= \mathbb{P}\left( \frac{w_1^* \sum_{i=1}^{d}(x_i - \mu_-)}{\sqrt{w_1^{*2} \sum_{i=1}^{d} \sigma_-^2}} > \frac{-b + dw_1^* \mu_-}{\sqrt{w_1^{*2} \sum_{i=1}^{d} \sigma_-^2}} \right)$$

$$= 1 - \Phi\left( \frac{-b + dw_1^* \mu_-}{\sqrt{d} w_1^* \sigma_-} \right)$$

where $\Phi$ is c.d.f. of normal Gaussian distribution $\mathcal{N}(0, 1)$. Then, we get

$$R_{nat}(f_{nat}) = \alpha \Phi\left( -\frac{b + dw_1^* \mu_+}{\sqrt{d} w_1^* \sigma_+} \right) + (1 - \alpha)\Phi\left( \frac{b - dw_1^* \mu_-}{\sqrt{d} w_1^* \sigma_-} \right).$$

For simplicity, we denote $\eta^* = \frac{b^*}{w_1^*}$. Then, we will find the optimal $\eta^*$ which minimize the overall natural risk $R_{nat}(f_{nat})$ by takeing $\frac{dR_{nat}(f_{nat})}{d\eta^*} = 0$. In detail, it is

$$\alpha \frac{1}{\sqrt{2\pi}} \exp(-\frac{1}{2}(\frac{\eta^* + d\mu_+}{\sqrt{d}\sigma_+})^2)\frac{-1}{\sqrt{d}\sigma_+} + (1 - \alpha)\frac{1}{\sqrt{2\pi}} \exp(-\frac{1}{2}(\frac{\eta^* - d\mu_-}{\sqrt{d}\sigma_-})^2)\frac{1}{\sqrt{d}\sigma_-} = 0. \quad (23)$$

Simplifying it gives

$$\left( \frac{\eta^* + d\mu_+}{\sigma_+} \right)^2 - \left( \frac{\eta^* - d\mu_-}{\sigma_-} \right)^2 = 2d\log\left( \frac{\alpha\sigma_-}{(1-\alpha)\sigma_+} \right). \quad (24)$$

Denote $K = d\log\left( \frac{\alpha\sigma_-}{(1-\alpha)\sigma_+} \right)$. Without loss of generality, we assume $K > 0$. Then we obtain

$$(\sigma_-^2 - \sigma_+^2)\eta^{*2} + 2d(\mu_+ \sigma_-^2 + \mu_- \sigma_+^2)\eta^* + d^2(\mu_+^2 \sigma_-^2 - \mu_-^2 \sigma_+^2) = 2K\sigma_+^2 \sigma_-^2. \quad (25)$$

Hence, $\eta^*$ can be expressed as follows:

$$\eta^* = \frac{-d(\mu_+ \sigma_-^2 + \mu_- \sigma_+^2) + \sigma_+ \sigma_-(\mu_+ + \mu_-)\sqrt{1 + 2K\frac{\sigma_-^2 - \sigma_+^2}{(\mu_+ + \mu_-)^2}}}{\sigma_-^2 - \sigma_+^2}. \quad (26)$$

Then class-wise natural risk is

$$R_{nat}^{+1}(f_{nat}) = \Phi\left( -\frac{\eta^* + d\mu_+}{\sqrt{d}\sigma_+} \right)$$

$$R_{nat}^{-1}(f_{nat}) = \Phi\left( \frac{\eta^* - d\mu_-}{\sqrt{d}\sigma_-} \right).$$

In particular, when $\sigma_+ = \sigma_- = \sigma$, then Equation (25) can be simplified as

$$2d(\mu_+ + \mu_-)\eta^* + d^2(\mu_+^2 - \mu_-^2) = 2K\sigma^2. \quad (27)$$

In this case, $\eta^*$ can be expressed as follows:

$$\eta^* = \frac{-d^2(\mu_+^2 - \mu_-^2) + 2K\sigma^2}{2d(\mu_+ + \mu_-)}. \quad (28)$$

and corresponding class-wise natural risk can be expressed as follows:

$$R_{nat}^{+1}(f_{nat}) = \Phi\left( \frac{-d^2(\mu_+ + \mu_-)^2 - 2K\sigma^2}{2d^{3/2}\sigma(\mu_+ + \mu_-)} \right)$$

$$R_{nat}^{-1}(f_{nat}) = \Phi\left( \frac{-d^2(\mu_+ + \mu_-)^2 + 2K\sigma^2}{2d^{3/2}\sigma(\mu_+ + \mu_-)} \right).$$

$\square$

## B.2 Adversarially Trained Linear model

**Theorem 5.3.** *For the adversarial trained linear classifier $f_{adv}$ that minimizes the adversarial risk:*

$$f_{adv}(x) = \arg\min_f \mathbb{E}_{(x,y)\sim\mathcal{D}^*}[\sup_{z\in\mathcal{U}(x,\epsilon)} (\mathbb{1}(f(z)\neq y)], \tag{29}$$

*if $\sigma_+ \neq \sigma_-$, the class-wise adversarial risk can be expressed as follows:*

$$R_{adv}^{+1}(f_{adv}) = \Phi\left(\frac{-\gamma^* - d(\mu_+ - \epsilon_{test})}{\sqrt{d}\sigma_+}\right) \tag{30}$$

$$R_{adv}^{-1}(f_{adv}) = \Phi\left(\frac{\gamma^* - d(\mu_- - \epsilon_{test})}{\sqrt{d}\sigma_-}\right) \tag{31}$$

*where $\Phi(\cdot)$ is the c.d.f. of standard Gaussian distribution $\mathcal{N}(0, 1)$ and*

$$\gamma^* = \frac{B + \sigma_+\sigma_-(\mu_+ + \mu_- - 2\epsilon_{train})\sqrt{1 + \frac{2K(\sigma_-^2 - \sigma_+^2)}{(\mu_+ + \mu_- - 2\epsilon_{train})^2}}}{\sigma_-^2 - \sigma_+^2}$$

*where $B = -d(\mu_+\sigma_-^2 + \mu_-\sigma_+^2 - \epsilon_{train}(\sigma_+^2 + \sigma_-^2))$ and $K$ is a positive constant. If $\sigma_+ = \sigma_- = \sigma$, the class-wise adversarial risk can be expressed as follows:*

$$R_{adv}^{+1}(f_{adv}) = \Phi\left(\frac{-C - 2K\sigma^2}{2d^{3/2}\sigma(\mu_+ + \mu_- - 2\epsilon_{train})}\right)$$

$$R_{adv}^{-1}(f_{adv}) = \Phi\left(\frac{-C + 2K\sigma^2}{2d^{3/2}\sigma(\mu_+ + \mu_- - 2\epsilon_{train})}\right)$$

*where $C = d^2(\mu_+ + \mu_- - 2\epsilon_{train})(\mu_+ + \mu_- - 2\epsilon_{test})$.*

*Proof.* For any classifier $f(x)$ in Equation (2), we first calculate its adversarial risk.

$$R_{adv}(f) = \mathbb{E}_{(x,y)\sim\mathcal{D}^*}\left[\sup_{z\in\mathcal{U}(x,\epsilon)} (\mathbb{1}(f(z)\neq y)\right]$$

$$= \alpha \cdot \mathbb{P}\left(\sup_{z\in\mathcal{U}(x,\epsilon)} \mathbb{1}(f(z) = -1)|y = +1\right) + (1-\alpha) \cdot \mathbb{P}\left(\sup_{z\in\mathcal{U}(x,\epsilon)} \mathbb{1}(f(z) = +1)|y = -1\right)$$

$$= \alpha R_{adv}^{+1}(f) + (1-\alpha)R_{adv}^{-1}(f).$$

Denote $f_{adv}(x) = \langle w^*, x\rangle + b^*$. Similar to Theorem 5.2, we can proof $w_1^* = w_2^* = \cdots = w_d^*$. Then $R_{adv}(f, +1)$ and $R_{adv}(f, -1)$ can be expressed as

$$R_{adv}^{+1}(f_{adv}) = \mathbb{P}(w_1^* \sum_{i=1}^d (x_i - \epsilon) + b < 0)$$

$$= \mathbb{P}\left(\frac{w_1^* \sum_{i=1}^d (x_i - \mu_+)}{\sqrt{w_1^{*2} \sum_{i=1}^d \sigma_+^2}} < \frac{-b - d(\mu_+ - \epsilon)w_1^*}{\sqrt{w_1^{*2} \sum_{i=1}^d \sigma_+^2}}\right)$$

$$= \Phi\left(-\frac{b + d(\mu_+ - \epsilon)w_1^*}{\sqrt{d}w_1^*\sigma_+}\right)$$

$$R_{adv}^{-1}(f_{adv}) = \mathbb{P}(w_1^* \sum_{i=1}^d (x_i + \epsilon) + b > 0)$$

$$= \mathbb{P}\left(\frac{w_1^* \sum_{i=1}^d (x_i + \mu_-)}{\sqrt{w_1^{*2} \sum_{i=1}^d \sigma_-^2}} > \frac{-b + d(\mu_- - \epsilon)w_1^*}{\sqrt{w_1^{*2} \sum_{i=1}^d \sigma_-^2}}\right)$$

$$= 1 - \Phi\left(\frac{-b + d(\mu_- - \epsilon)w_1^*\mu_-}{\sqrt{d}w_1^*\sigma_-}\right).$$

Then, we get

$$R_{adv}(f_{adv}) = \alpha\Phi\left(-\frac{b + d(\mu_+ - \epsilon)w_1^*}{\sqrt{d}w_1^*\sigma_+}\right) + (1-\alpha)\Phi\left(\frac{b - d(\mu_- - \epsilon)w_1^*\mu_-}{\sqrt{d}w_1^*\sigma_-}\right). \tag{32}$$

We denote by $\epsilon_{train}$ and $\epsilon_{test}$ perturbation radii in training phase and testing phase, respectively. For simplicity, we denote $\gamma^* = \frac{b^*}{w_1^*}$. Now, we will find the optimal $\gamma^*$ which minimizes adversarial risk $R_{adv}(f_{f_{adv}})$ by takeing $\frac{dR_{adv}(f_{adv})}{d\gamma^*} = 0$. In detail, it is

$$\frac{\alpha}{\sqrt{2\pi}} \exp(-\frac{1}{2}(\frac{\gamma^* + d(\mu_+ - \epsilon_{train})}{\sqrt{d}\sigma_+})^2)\frac{-1}{\sqrt{d}\sigma_+} + \frac{1-\alpha}{\sqrt{2\pi}} \exp(-\frac{1}{2}(\frac{\gamma^* - d(\mu_- - \epsilon_{train})}{\sqrt{d}\sigma_-})^2)\frac{1}{\sqrt{d}\sigma_-} = 0.$$

Simplifying it gives

$$\left(\frac{\gamma^* + d(\mu_+ - \epsilon_{train})}{\sigma_+}\right)^2 - \left(\frac{\gamma^* - d(\mu_- - \epsilon_{train})}{\sigma_-}\right)^2 = 2d\log\left(\frac{\alpha\sigma_-}{(1-\alpha)\sigma_+}\right) = 2K.$$

Hence, $\gamma^*$ can be expressed as follows:

$$\gamma^* = \frac{-d(\mu_+\sigma_-^2 + \mu_-\sigma_+^2 - \epsilon_{train}(\sigma_+^2 + \sigma_-^2)) + \sigma_+\sigma_-(\mu_+ + \mu_- - 2\epsilon_{train})\sqrt{1 + 2K\frac{\sigma_-^2 - \sigma_+^2}{(\mu_+ + \mu_- - 2\epsilon_{train})^2}}}{\sigma_-^2 - \sigma_+^2}.$$

Then class-wise robust risk is

$$R_{adv}^{+1}(f_{adv}) = \Phi\left(-\frac{\gamma^* + d(\mu_+ - \epsilon_{test})}{\sqrt{d}\sigma_+}\right)$$

$$R_{adv}^{-1}(f_{adv}) = \Phi\left(\frac{\gamma^* - d(\mu_- - \epsilon_{test})}{\sqrt{d}\sigma_-}\right).$$

In particular, when $\sigma_+ = \sigma_- = \sigma$, then we have

$$\gamma^* = \frac{2K\sigma^2 - d^2((\mu_+ - \epsilon_{train})^2 - (\mu_- - \epsilon_{train})^2)}{2d(\mu_+ + \mu_- - 2\epsilon_{train})} \tag{33}$$

and corresponding robust risk is

$$R_{adv}^{+1}(f_{adv}) = \Phi\left(-\frac{d^2(\mu_+ + \mu_- - 2\epsilon_{train})(\mu_+ + \mu_- - 2\epsilon_{test}) + 2K\sigma^2}{2d^{3/2}\sigma(\mu_+ + \mu_- - 2\epsilon_{train})}\right)$$

$$R_{adv}^{-1}(f_{adv}) = \Phi\left(\frac{-d^2(\mu_+ + \mu_- - 2\epsilon_{train})(\mu_+ + \mu_- - 2\epsilon_{test}) + 2K\sigma^2}{2d^{3/2}\sigma(\mu_+ + \mu_- - 2\epsilon_{train})}\right).$$

$\square$

## B.3 Tradeoff between Average Robustness and Robust Fairness

**Theorem 5.5.** *Given an adversarially trained linear model $f_{adv}$ in Equation (7), the variance of class-wise robust accuracy $VCRA(f_{adv})$ is increasing with respect to $\epsilon_{train}$ if $\sigma_- \geq \sigma_+$.*

*Proof.* According to the results of Theorem 5.3, we can get class-wise robust accuracy:

$$p_{adv}(1) = 1 - R_{adv}^{+1}(f_{adv}) \tag{34}$$

$$p_{adv}(-1) = 1 - R_{adv}^{-1}(f_{adv}). \tag{35}$$

Then the variance of class-wise robust accuracy can be expressed as

$$VCRA(f_{adv}) = \text{Var}(p_{adv}(+1), p_{adv}(-1)) \tag{36}$$

$$= \text{Var}\left(1 - R_{adv}^{+1}(f_{adv}), 1 - R_{adv}^{-1}(f_{adv})\right) \tag{37}$$

$$= \text{Var}\left(R_{adv}^{+1}(f_{adv}), R_{adv}^{-1}(f_{adv})\right) \tag{38}$$

$$= \frac{\left(R_{adv}^{+1}(f_{adv}) - R_{adv}^{-1}(f_{adv})\right)^2}{2}. \tag{39}$$

If $\sigma_- \geq \sigma_+$, we can easily verify that $R_{adv}^{+1}(f_{adv})$ is decreasing and $R_{adv}^{-1}(f_{adv})$ is increasing with respect to $\epsilon_{train}$ according to Theorem 5.3. Hence, $VCRA(f_{adv})$ is increasing with respect to $\epsilon_{train}$. $\square$

**Theorem 5.6.** *Given an adversarially trained linear model $f_{adv}$ in Equation (7) and suppose $\sigma_+ = \sigma_-$. The variance of class-wise robust accuracy of $f_{adv}$ can be expressed as follows:*

$$VCRA(f_{adv}) = \frac{1}{\pi d^3} \exp(-\xi^2)\left(\frac{K\sigma}{\mu_+ + \mu_- - 2\epsilon_{train}}\right)^2$$

*where $\xi$ is a constant and $K$ is a positive constant.*

*Proof.* According to Lagrange's Mean Value Theorem, there exists some $\xi$ such that

$$R_{adv}^{-1}(f_{adv}) - R_{adv}^{+1}(f_{adv}) \tag{40}$$

$$= \Phi\left(\frac{-d^2(\mu_+ + \mu_- - 2\epsilon_{train})(\mu_+ + \mu_- - 2\epsilon_{test}) + 2K\sigma^2}{2d^{3/2}\sigma(\mu_+ + \mu_- - 2\epsilon_{train})}\right) \tag{41}$$

$$- \Phi\left(-\frac{d^2(\mu_+ + \mu_- - 2\epsilon_{train})(\mu_+ + \mu_- - 2\epsilon_{test}) + 2K\sigma^2}{2d^{3/2}\sigma(\mu_+ + \mu_- - 2\epsilon_{train})}\right) \tag{42}$$

$$= \Phi'(\xi)\left(\frac{-C + 2K\sigma^2}{2d^{3/2}\sigma(\mu_+ + \mu_- - 2\epsilon_{train})} + \frac{C + 2K\sigma^2}{2d^{3/2}\sigma(\mu_+ + \mu_- - 2\epsilon_{train})}\right) \tag{43}$$

$$= \frac{1}{\sqrt{2\pi}}\exp(-\frac{\xi^2}{2})\left(\frac{2K\sigma}{d^{3/2}(\mu_+ + \mu_- - 2\epsilon_{train})}\right) \tag{44}$$

where $C = d^2(\mu_+ + \mu_- - 2\epsilon_{train})(\mu_+ + \mu_- - 2\epsilon_{test})$. Therefore, we obtain final expression of variance:

$$VCRA(f_{adv}) = \frac{1}{\pi d^3}\exp(-\xi^2)\left(\frac{K\sigma}{\mu_+ + \mu_- - 2\epsilon_{train}}\right)^2. \tag{45}$$

$\square$

**Theorem 5.7.** *Given an adversarially trained linear model $f_{adv}$ in Equation (7) and a naturally trained linear model $f_{nat}$ in Equation (3), and suppose $\sigma_+ = \sigma_-$. If $\epsilon_{train} < \frac{\mu_+ + \mu_-}{e}$ and $\epsilon_{test} < \frac{\mu_+ + \mu_-}{2} - \frac{eK^3\sigma^3}{d^4(\mu_+ + \mu_-)}$, we have*

$$\frac{VCRA(f_{adv})}{VCRA(f_{nat})} = \exp(\zeta_2^2 - \zeta_1^2)\cdot\left(\frac{\mu_+ + \mu_-}{\mu_+ + \mu_- - 2\epsilon_{train}}\right)^2 \geq 1$$

*where $K$ is a positive constant, $\zeta_1$, $\zeta_2$ are two constants.*

*Proof.* According to Theorem 5.2 and Theorem 5.3, we have

$$R_{adv}^{+1}(f_{nat}) = \Phi\left(-\frac{d^2(\mu_+ + \mu_-)(\mu_+ + \mu_- - 2\epsilon_{test}) + 2K\sigma^2}{2d^{3/2}\sigma(\mu_+ + \mu_- - 2\epsilon_{train})}\right)$$

$$R_{adv}^{-1}(f_{nat}) = \Phi\left(\frac{-d^2(\mu_+ + \mu_-)(\mu_+ + \mu_- - 2\epsilon_{test}) + 2K\sigma^2}{2d^{3/2}\sigma(\mu_+ + \mu_- - 2\epsilon_{train})}\right).$$

Denote $C = d^2(\mu_+ + \mu_- - 2\epsilon_{train})(\mu_+ + \mu_- - 2\epsilon_{test})$. Similar to the proof of Theorem 5.6, according to Lagrange Mean Value Theorem, there exits $\zeta_1$ and $\zeta_2$ such that

$$R_{adv}^{-1}(f_{adv}) - R_{adv}^{+1}(f_{adv}) = \frac{1}{\sqrt{2\pi}}\exp(-\frac{\zeta_1^1}{2})\left(\frac{2K\sigma}{d^{3/2}(\mu_+ + \mu_- - 2\epsilon_{train})}\right) \tag{46}$$

and

$$R_{adv}^{-1}(f_{nat}) - R_{adv}^{+1}(f_{nat}) = \frac{1}{\sqrt{2\pi}}\exp(-\frac{\zeta_2^2}{2})\left(\frac{2K\sigma}{d^{3/2}(\mu_+ + \mu_-)}\right) \tag{47}$$

where

$$\zeta_1 \in \left(-\frac{C + 2K\sigma^2}{2d^{3/2}\sigma(\mu_+ + \mu_- - 2\epsilon_{train})}, \frac{-C + 2K\sigma^2}{2d^{3/2}\sigma(\mu_+ + \mu_- - 2\epsilon_{train})}\right) \tag{48}$$

$$\zeta_2 \in \left(-\frac{d^2(\mu_+ + \mu_-)(\mu_+ + \mu_- - 2\epsilon_{test}) + 2K\sigma^2}{2d^{3/2}\sigma(\mu_+ + \mu_-)}, \frac{-d^2(\mu_+ + \mu_-)(\mu_+ + \mu_- - 2\epsilon_{test}) + 2K\sigma^2}{2d^{3/2}\sigma(\mu_+ + \mu_-)}\right) \tag{49}$$

Therefore, we get

$$\frac{VCRA(f_{adv})}{VCRA(f_{nat})} = \frac{\frac{1}{\pi d^3}\exp(-\zeta_1^2)\left(\frac{K\sigma}{d^{3/2}(\mu_+ + \mu_- - 2\epsilon)}\right)^2}{\frac{1}{\pi d^3}\exp(-\zeta_2^2)\left(\frac{K\sigma}{d^{3/2}(\mu_+ + \mu_-)}\right)^2} \tag{50}$$

$$= \exp(\zeta_2^2 - \zeta_1^2)\cdot\left(\frac{\mu_+ + \mu_-}{\mu_+ + \mu_- - 2\epsilon_{train}}\right)^2. \tag{51}$$

If $\epsilon_{train} < \frac{\mu_+ + \mu_-}{e}$ and $\epsilon_{test} < \frac{\mu_+ + \mu_-}{2} - \frac{eK^3\sigma^3}{d^4(\mu_+ + \mu_-)}$, we can get

$$\frac{VCRA(f_{adv})}{VCRA(f_{nat})} \geq \exp\left(\frac{-4K^3\sigma^3}{d^4(\mu_+ + \mu_- - 2\epsilon_{test})(\mu_+ + \mu_- - 2\epsilon_{train})}\right)\cdot\left(\frac{\mu_+ + \mu_-}{\mu_+ + \mu_- - 2\epsilon_{train}}\right)^2 \geq 1. \tag{52}$$

$\square$

**Theorem B.6.** *Given an adversarial trained linear model $f_{adv}$ in Equation (7) and a natural trained linear model $f_{nat}$ in Equation (3),and suppose $\sigma_+ = \sigma_-$. If $\epsilon_{test} < \sqrt{\frac{\mu_+ + \mu_-}{2} - \frac{2K\sigma^2}{d^2(\mu_+ + \mu_-)}}$ and $\epsilon_{train} < \frac{\mu_+ + \mu_-}{2} - \frac{K\sigma}{\epsilon_{test} d^{3/2}}$, we have*

$$\frac{VCRA(f_{adv})}{VCA(f_{nat})} = \exp(\xi_2^2 - \xi_1^2) \cdot \left( \frac{\mu_+ + \mu_-}{\mu_+ + \mu_- - 2\epsilon_{train}} \right)^2 \geq 1$$

*where $K$ is a positive constant, $\xi_1$, $\xi_2$ are two constants and $\xi_2 < \xi_1 < 0$,*

*Proof.* Under the conditions that $\epsilon_{train} < \frac{\mu_+ + \mu_-}{2} - \frac{K\sigma}{\epsilon_{test} d^{3/2}}$ and $\epsilon_{test} < \sqrt{\frac{\mu_+ + \mu_-}{2} - \frac{2K\sigma^2}{d^2(\mu_+ + \mu_-)}}$, we obtain following relations:

$$-\frac{d^2(\mu_+ + \mu_-)^2 + 2K\sigma^2}{2d^{3/2}\sigma(\mu_+ + \mu_-)} < \frac{-d^2(\mu_+ + \mu_-)^2 + 2K\sigma^2}{2d^{3/2}\sigma(\mu_+ + \mu_-)} <$$

$$\frac{-d^2(\mu_+ + \mu_- - 2\epsilon_{train})(\mu_+ + \mu_- - 2\epsilon_{test}) - 2K\sigma^2}{2d^{3/2}\sigma(\mu_+ + \mu_- - 2\epsilon_{train})}$$

$$< \frac{-d^2(\mu_+ + \mu_- - 2\epsilon_{train})(\mu_+ + \mu_- - 2\epsilon_{test}) + 2K\sigma^2}{2d^{3/2}\sigma(\mu_+ + \mu_- - 2\epsilon_{train})} < 0.$$

Denote $C = d^2(\mu_+ + \mu_- - 2\epsilon_{train})(\mu_+ + \mu_- - 2\epsilon_{test})$. Similar to the proof of Theorem 5.7, according to Lagrange Mean Value Theorem, there exits $\xi_1$ and $\xi_2$ such that

$$R_{adv}^{-1}(f_{adv}) - R_{adv}^{+1}(f_{adv}) = \tag{53}$$

$$= \frac{1}{\sqrt{2\pi}} \exp(-\frac{\xi_1^1}{2}) \left( \frac{2K\sigma}{d^{3/2}(\mu_+ + \mu_- - 2\epsilon_{train})} \right) \tag{54}$$

and

$$R_{nat}^{-1}(f_{nat}) - R_{nat}^{+1}(f_{nat}) = \tag{55}$$

$$\Phi'(\xi_2) \left( \frac{-d^2(\mu_+ + \mu_-)^2 + 2K\sigma^2}{2d^{3/2}\sigma(\mu_+ + \mu_-)} + \frac{d^2(\mu_+ + \mu_-)^2 + 2K\sigma^2}{2d^{3/2}\sigma(\mu_+ + \mu_-)} \right) \tag{56}$$

$$\frac{1}{\sqrt{2\pi}} \exp(-\frac{\xi_2^2}{2}) \left( \frac{2K\sigma}{d^{3/2}(\mu_+ + \mu_-)} \right) \tag{57}$$

where

$$\xi_1 \in \left( -\frac{C + 2K\sigma^2}{2d^{3/2}\sigma(\mu_+ + \mu_- - 2\epsilon_{train})}, \frac{-C + 2K\sigma^2}{2d^{3/2}\sigma(\mu_+ + \mu_- - 2\epsilon_{train})} \right) \tag{58}$$

$$\xi_2 \in \left( -\frac{d^2(\mu_+ + \mu_-)^2 + 2K\sigma^2}{2d^{3/2}\sigma(\mu_+ + \mu_-)}, \frac{-d^2(\mu_+ + \mu_-)^2 + 2K\sigma^2}{2d^{3/2}\sigma(\mu_+ + \mu_-)} \right) \tag{59}$$

Note that Equation (53) implies $\xi_2 < \xi_1 < 0$ and thus $\xi_1^2 < \xi_2^2$. Then we get

$$\frac{\text{VCRA}(\text{f}_{adv})}{\text{VCA}(\text{f}_{nat})} = \frac{\frac{1}{\pi d^3} \exp(-\xi_1^2) \left( \frac{K\sigma}{d^{3/2}(\mu_+ + \mu_- - 2\epsilon)} \right)^2}{\frac{1}{\pi d^3} \exp(-\xi_2^2) \left( \frac{K\sigma}{d^{3/2}(\mu_+ + \mu_-)} \right)^2} \tag{60}$$

$$= \exp(\xi_2^2 - \xi_1^2) \cdot \left( \frac{\mu_+ + \mu_-}{\mu_+ + \mu_- - 2\epsilon_{train}} \right)^2 \geq 1. \tag{61}$$

$\square$

## B.4 Fairly Adversarial Training

**Theorem B.7.** *[1]. Let $(x_1, y_1)$, $(x_2, y_2)$, ...,$(x_n, y_n)$ be drawn independent and identically distributed (i.i.d.) from the unknown distribution $\mathcal{D}$. Under appropriate conditions on the loss $\ell(\cdot)$, parameter space $\Theta$, with probability of at least $1 - \delta$, the following holds for all $\theta \in \Theta$,*

$$R_{nat}(f) \leq \hat{R}_{nat}(f) + \sqrt{\frac{\text{Var}\,\ell(\theta, x, y)\frac{1}{\delta}}{n}} + \frac{C}{n} \tag{62}$$

*where $C$ is a constant, $\hat{R}_{nat}(f) = \frac{1}{n} \sum_{i=1}^{n} \ell(\theta, x_i, y_i)$ and $\text{Var}$ indicates the variance.*

**Theorem 6.1.** *Under the conditions of Theorem B.7, with probability of at least $1 - \delta$, the following holds for all $\theta \in \Theta$:*

$$R_{adv}(f) \leq +\hat{R}_{adv}(f) + \sqrt{\frac{\mathrm{VCAR}(f)\frac{1}{\delta}}{n}} + \frac{C}{n}. \tag{63}$$

*Proof.* For simplicity, we denote $\bar{\ell}(\theta, x, y) = \frac{1}{L} \sum_{j=1}^{L} \ell(\theta, x, i)$. According to the properties of conditional expectation, we have

$$
\begin{aligned}
\mathrm{Var}(\ell(\theta, x, y)) &= \mathbb{E}_{x \sim \mathcal{D}_x} \left( \mathrm{Var}(\ell(\theta, x, y) | x) \right) \\
&= \mathbb{E}_{x \sim \mathcal{D}_x} \left[ \frac{1}{L} \sum_{i=1}^{L} (\ell(\theta, x, i) - \bar{\ell}(\theta, x, y))^2 \right] \\
&= \frac{1}{L} \sum_{i=1}^{L} \mathbb{E}_{x \sim \mathcal{D}_x} \left[ (\ell(\theta, x, i) - \bar{\ell}(\theta, x, y))^2 \right].
\end{aligned} \tag{64}
$$

Because squared function is convex, according to Jensen's inequality, we have

$$
\begin{aligned}
\mathbb{E}_{x \sim \mathcal{D}_x} \left[ (\ell(\theta, x, i) - \bar{\ell}(\theta, x, y))^2 \right] &\leq \left[ \mathbb{E}_{x \sim \mathcal{D}_x} (\ell(\theta, x, i) - \bar{\ell}(\theta, x, y)) \right]^2 \\
&= \left[ \mathbb{E}_{x \sim \mathcal{D}_x} (\ell(\theta, x, i)) - \frac{1}{L} \sum_{j=1}^{L} \mathbb{E}_{x \sim \mathcal{D}_x} \ell(\theta, x, j)) \right]^2 \\
&= \left[ R(f, i) - \frac{1}{L} \sum_{j=1}^{L} R(f, j) \right]^2.
\end{aligned} \tag{65}
$$

Similar to [27], we replace $\ell(\theta, x, y)$ with $\tilde{\ell}(\theta, x, y)$ in Equation (65). We can then obtain the following inequality:

$$\mathrm{Var}(\tilde{\ell}(\theta, x, y)) \leq \mathrm{VCAR}(f). \tag{66}$$

Combining with Theorem B.7, we get

$$R_{adv}(f) \leq \frac{1}{n} \sum_{i=1}^{n} \tilde{\ell}(\theta, x_i, y_i) + \sqrt{\frac{\mathrm{VCAR}(f)\frac{1}{\delta}}{n}}) + \frac{C}{n}. \tag{67}$$

$\square$

## C   More Experimental Results in Section 7

### C.1   More Experimental Results of FAT

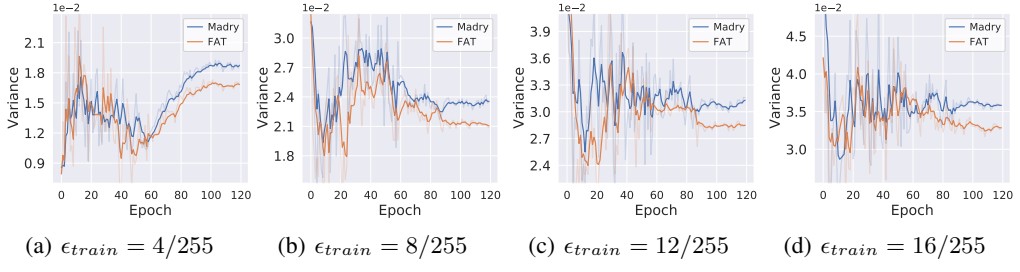

(a) $\epsilon_{train} = 4/255$     (b) $\epsilon_{train} = 8/255$     (c) $\epsilon_{train} = 12/255$     (d) $\epsilon_{train} = 16/255$

Figure 3: The variance of class-wise robust accuracy for our proposed FAT and Madry using ResNet18 on CIFAR-10. The perturbation radius for AT is chosen from $\epsilon_{train} = \{4/255, 8/255, 12/255, 16/255\}$. The adversarial testing examples are generated by FGSM with testing perturbation radius $\epsilon_{test} = 16/255$.

In Figures 3, 4 and 5, Tables 8, 9, 10 and 11, we present more experimental results to validate the effectiveness of FAT under ResNet18. The implementation details resemble those in Section 7. From the results above, we can make similar observations to these in Section 7. Specifically, compared with Madry and TRADES, FAT obtains smaller variance of class-wise robust accuracy while achieving comparable average robust accuracy.

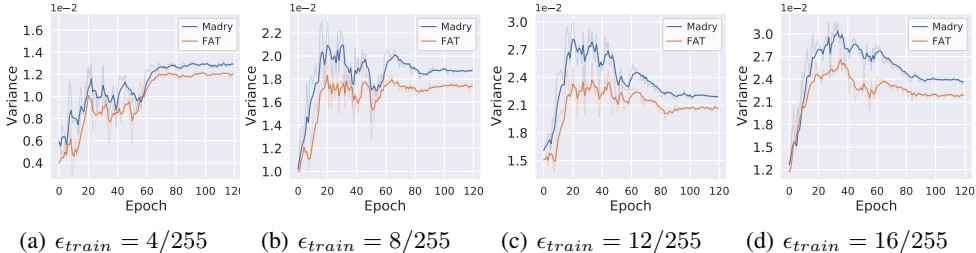

(a) $\epsilon_{train} = 4/255$  (b) $\epsilon_{train} = 8/255$  (c) $\epsilon_{train} = 12/255$  (d) $\epsilon_{train} = 16/255$

Figure 4: The variance of class-wise robust accuracy for our proposed FAT and Madry using ResNet18 on CIFAR-100. The perturbation radii for AT are chosen from $\epsilon_{train} = \{4/255, 8/255, 12/255, 16/255\}$. The adversarial testing examples are generated by FGSM with testing perturbation radius $\epsilon_{test} = 16/255$.

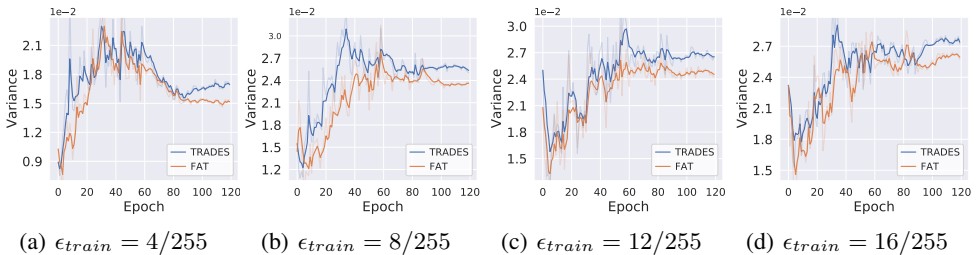

(a) $\epsilon_{train} = 4/255$  (b) $\epsilon_{train} = 8/255$  (c) $\epsilon_{train} = 12/255$  (d) $\epsilon_{train} = 16/255$

Figure 5: The variance of class-wise robust accuracy for our proposed FAT and TRADES using ResNet18 on CIFAR-10. The perturbation radii for AT are chosen from $\epsilon_{train} = \{4/255, 8/255, 12/255, 16/255\}$. The adversarial testing examples are generated by FGSM with testing perturbation radius $\epsilon_{test} = 16/255$.

Table 8: The variance of class-wise robust accuracy $(10^{-2})$ for Madry and FAT using ResNet18 on CIFAR-100. The perturbation radii are chosen from $\epsilon_{train} = \{4/255, 8/255, 12/255, 16/255\}$. The adversarial testing examples are generated by FGSM, PGD-20 and C&W with different testing perturbation radii $\epsilon_{test} = \{16/255, 20/255\}$.

| attack algorithm | FGSM | | | | PGD-20 | | | | C&W | | | |
|---|---|---|---|---|---|---|---|---|---|---|---|---|
| $\epsilon_{test}$ | 16/255 | | 20/255 | | 16/255 | | 20/255 | | 16/255 | | 20/255 | |
| $\epsilon_{train}$ | PGD | FAT | PGD | FAT | PGD | FAT | PGD | FAT | PGD | FAC | PDG | FAT |
| 4/255 | 1.25 | 1.25 | 0.81 | 0.79 | 0.19 | 0.19 | 0.05 | 0.05 | 0.23 | 0.22 | 0.07 | 0.06 |
| 8/255 | 1.80 | 1.69 | 1.31 | 1.22 | 0.60 | 0.57 | 0.26 | 0.25 | 0.70 | 0.66 | 0.33 | 0.31 |
| 12/255 | 2.17 | 2.04 | 1.60 | 1.47 | 0.96 | 0.95 | 0.52 | 0.44 | 0.97 | 0.95 | 0.54 | 0.47 |
| 16/255 | 2.32 | 2.17 | 1.89 | 1.77 | 1.12 | 1.08 | 0.65 | 0.60 | 1.18 | 1.09 | 0.65 | 0.61 |

Table 9: The average robust accuracy $(10^{-2})$ for Madry and FAT using ResNet18 on CIFAR-100. The perturbation radii are chosen from $\epsilon_{train} = \{4/255, 8/255, 12/255, 16/255\}$. The adversarial testing examples are generated by FGSM, PGD-20 and C&W with different testing perturbation radii $\epsilon_{test} = \{16/255, 20/255\}$. We use **bold** to denote higher robust accuracy.

| attack algorithm | FGSM | | | | PGD-20 | | | | C&W | | | |
|---|---|---|---|---|---|---|---|---|---|---|---|---|
| $\epsilon_{test}$ | 16/255 | | 20/255 | | 16/255 | | 20/255 | | 16/255 | | 20/255 | |
| $\epsilon_{train}$ | PGD | FAT | PGD | FAT | PGD | FAT | PGD | FAT | PGD | FAC | PDG | FAT |
| 4/255 | 11.94 | 11.89 | 8.01 | 7.89 | 1.83 | **2.00** | 0.69 | **0.78** | 2.15 | 2.14 | 0.76 | **0.83** |
| 8/255 | 16.52 | 16.41 | 12.35 | 11.94 | 5.16 | **5.19** | 2.65 | 2.62 | 5.65 | 5.49 | 3.10 | 2.87 |
| 12/255 | 18.56 | **18.74** | 14.76 | **14.79** | 7.81 | **7.82** | 4.43 | 4.26 | 8.06 | 8.01 | 4.58 | **4.61** |
| 16/255 | 19.78 | **19.81** | 16.26 | **16.78** | 8.95 | **9.01** | 5.42 | **5.47** | 9.03 | **9.53** | 5.54 | **5.62** |

Moreover, this finding still holds under various neural networks, attack algorithms and datasets. These observations suggest that FAT can effectively suppress the growth of variance of class-wise robust accuracy while improving average robust accuracy compared with Madry and TRADES. Thus, FAT can mitigate the tradeoff between robustness and robust fairness.

Table 10: The variance of class-wise robust accuracy $(10^{-2})$ for TRADES and FAT using ResNet18 on CIFAR-10. The perturbation radii are chosen from $\epsilon_{train} = \{4/255, 8/255, 12/255, 16/255\}$. The adversarial testing examples are generated by FGSM, PGD-20 and C&W with different testing perturbation radii $\epsilon_{test} = \{16/255, 20/255\}$.

| attack algorithm | FGSM | | | | PGD-20 | | | | C&W | | | |
|---|---|---|---|---|---|---|---|---|---|---|---|---|
| $\epsilon_{test}$ | 16/255 | | 20/255 | | 16/255 | | 20/255 | | 16/255 | | 20/255 | |
| $\epsilon_{train}$ | TRADES | FAT | TRADES | FAT | TRADES | FAT | TRADES | FAT | TRADES | FAC | TRADES | FAT |
| 4/255 | 1.59 | 1.50 | 1.15 | 1.08 | 0.18 | 0.15 | 0.04 | 0.03 | 0.19 | 0.17 | 0.04 | 0.04 |
| 8/255 | 2.48 | 2.45 | 2.18 | 2.09 | 0.73 | 0.65 | 0.25 | 0.23 | 0.76 | 0.76 | 0.28 | 0.21 |
| 12/255 | 2.64 | 2.48 | 2.45 | 2.39 | 1.48 | 1.46 | 0.71 | 0.65 | 1.37 | 1.31 | 0.62 | 0.59 |
| 16/255 | 2.69 | 2.64 | 2.50 | 2.45 | 1.67 | 1.59 | 0.84 | 0.73 | 1.55 | 1.47 | 0.73 | 0.71 |

Table 11: The average robust accuracy $(10^{-2})$ for TRADES and FAT using ResNet18 on CIFAR-10. The perturbation radii are chosen from $\epsilon_{train} = \{4/255, 8/255, 12/255, 16/255\}$. The adversarial testing examples are generated by FGSM, PGD-20 and C&W with different testing perturbation radii $\epsilon_{test} = \{16/255, 20/255\}$. We use **bold** to denote higher robust accuracy.

| attack algorithm | FGSM | | | | PGD-20 | | | | C&W | | | |
|---|---|---|---|---|---|---|---|---|---|---|---|---|
| $\epsilon_{test}$ | 16/255 | | 20/255 | | 16/255 | | 20/255 | | 16/255 | | 20/255 | |
| $\epsilon_{train}$ | TRADES | FAT | TRADES | FAT | TRADES | FAT | TRADES | FAT | TRADES | FAC | TRADES | FAT |
| 4/255 | 32.10 | **32.79** | 23.61 | **24.02** | 4.9 | **5.14** | 1.80 | **1.93** | 5.58 | 5.56 | 2.07 | 2.07 |
| 8/255 | 42.27 | **42.69** | 35.37 | 35.37 | 16.84 | **18.54** | 8.38 | **9.04** | 15.74 | 16.56 | 7.92 | **8.22** |
| 12/255 | 42.74 | **43.32** | 37.24 | **37.88** | 23.90 | 23.81 | 14.68 | 14.22 | 20.34 | **20.71** | 12.27 | **12.35** |
| 16/255 | 43.71 | **43.98** | 38.34 | **38.35** | 26.11 | **26.81** | 16.55 | **16.66** | 22.37 | **22.93** | 13.48 | **13.69** |

## C.2 Sensitivity of regularization hyperparameter

The regularization parameter $\lambda$ is an important hyperparameter in our proposed method. We show how the regularization parameter affects the performance of our robust classifiers by numerical experiments on CIFAR-10. From Figure 6, we can observe that as the regularization parameter $\lambda$ increases, the variance of class-wise robust accuracy decreases; while the average robust accuracy first increases and then decreases. Empirically, when we set the hyperparameter $\lambda = 0.1$, our method is able to learn classifiers with comparable average robust accuracy and smaller variance of class-wise robust accuracy compared with Madry and TRADES.

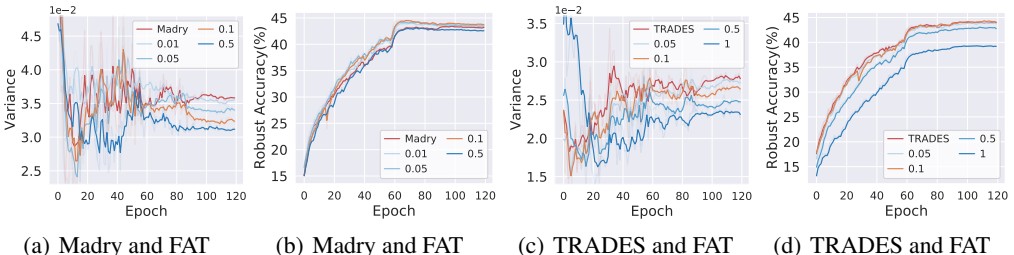

(a) Madry and FAT   (b) Madry and FAT   (c) TRADES and FAT   (d) TRADES and FAT

Figure 6: Sensitivity of regularization hyperparameter $\lambda$ on CIRAR-10.