# OpenReview forum: "On the Tradeoff Between Robustness and Fairness"
_NeurIPS.cc/2022/Conference — NeurIPS 2022 Accept_

### Official Review · Reviewer_8YA2 · 2022-07-09

**Rating:** 6
**Confidence:** 4
**Soundness:** 3 good
**Presentation:** 3 good
**Contribution:** 3 good

**Summary:**

This paper studies the relation between the variance of class-wise robust accuracy and perturbation radius in adversarial training. It finds that a tradeoff exists between average robustness and robust fairness. Theoretically, this paper analyzes this phenomenon of the tradeoff and provides an understanding. Finally, this paper proposes a variance based regularized method to mitigate the tradeoff. The idea of this paper is direct and reasonable, and the experimental results demonstrate the effectiveness of the proposed method.

**Questions:**

The theorems in this paper built on a simple linear model. How to extend them to DNN model with multi classification?

For the Theorem 4.5, does the situation σ_+ 〖>σ〗_- equal to the situation σ_+ 〖<σ〗_-? How the variance VCRA changes if σ_+ 〖>σ〗_-?

Can the methods in [1][2] also mitigate the variance of class-wise robust accuracy?

[1] Han Xu, Xiaorui Liu, Yaxin Li, Anil K. Jain, and Jiliang Tang. To be robust or to be fair: Towards fairness in adversarial training. In ICML, 2021.
[2] Qi Tian, Kun Kuang, Kelu Jiang, Fei Wu, and Yisen Wang. Analysis and applications of class-wise robustness in adversarial training. In KDD, 2021.


**Limitations:**

N.A

**Strengths And Weaknesses:**

Strengths:

1.	The paper is clearly written and is easy to follow.

2.	This paper reveals fairness of robust accuracy among different classes from a theoretical point of view.

3.	Theoretical proof and analysis are sufficient.

4.	Extensive experiments are shown to demonstrate the effectiveness of the proposed framework.

Weakness:

1.	Figures 1 (a), (b) and figure 2 (a), (b) are incomplete, and some contents are not displayed well.

3.	There are some grammatical errors, like “a understanding …” in line 34.

4.	If I understand correctly, the “sign(t)” should take “-1” for “otherwise” rather than “0” in line 145.

5.	For the equation between line 245 and 246 in Theorem 5.1, there is an extra plus sign.

6.	I am not sure that if Φ(·) in Theorem 4.2 is the probability distribution function. In the line 419 in appendix, the authors describe it as a cumulative distribution function.

7.	There are some typos in appendix: “P(f(x) = +1|y = +1)” in equation 19 should be “P(f(x) = +1|y = -1)”, “Rnat (f,1)” in line 411 should be “Rnat (f,-1 )”.

---

> ### Author Response · Authors · 2022-08-02
> **Response to Reviewer 8YA2**
>
> Dear Reviewer 8YA2,
>
> We are grateful to you for your encouraging comments and constructive suggestions on our work. We provide answers to the raised questions/cons below.
>
> **(Q8): Some miscellaneous minor issues about Figures, grammar and others.**
>
> We have fixed some minor issues in new revision, including:
>
> 1): Figure 1 (a), (b) and Figure 2 (a), (b) can be displayed well;
>
> 2): “an understanding” in line 41;
>
> 3): $sign(t)$ takes “-1” for otherwise in line 146;
>
> 4): delete an extra plus sign “+” between line 246 and 247;
>
> 5): $\Phi(\cdot)$ is cumulative distribution function in Theorem 5.2. We have described $\Phi(\cdot)$ as cumulative distribution function(c.d.f.) in full paper.
>
> 6): Some other typos in Appendix.
>
> **(Q9). How to extend the theorems in this paper to DNN model with multi classification?**
>
> Our theoretical results can be easily extended to two-layer neural network with ReLU activation function for binary classification with same proof techniques. Then, based on the strategy of one vs all, we can extend the theorems to multiclass classification. The goal of theoretical analysis is to provide a solid explanation for experimental phenomenon and reveal potential reasons behind it. We need explicit analytical solutions for optimization problems and thereby we choose linear model to explain why adversarial training leads to severer robust fairness issue.
>
> **(Q10): does the situation $\sigma_{+} > \sigma_{-}$ equal to the situation $\sigma_{+} < \sigma_{-}$ ?  How the variance VCRA changes if $\sigma_{+} > \sigma_{-}$ ?**
>
> Similar to the situation where $\sigma_{+} < \sigma_{-}$, $\sigma_{+} > \sigma_{-}$ means that class “+1” is more vulnerable than class “-1”. We can easily obtain similar theoretical results with a small modification of assumption (K < 0) and same proof, and therefore the variance VCRA is still increasing with respect to $\epsilon_{train}$ if $\sigma_{+} > \sigma_{-}$.
>
> **(Q11): Can the methods in [1][2] also mitigate the variance of class-wise robust accuracy?**
>
> The methods in [1][2] focus on improve the worst-class robust accuracy. Usually they can not guarantee a smaller variance of class-wise robust accuracy.

---

> ### Author Response · Authors · 2022-08-08
> **Discussion**
>
> Dear reviewer 8YA2,
>
> we will really appreciate it if the reviewer can go over our detailed response and revisions. Please feel free to ask us any questions you may still have and we will be more than happy to answer them.
>
> Thank you again for reviewing our paper and we look forward to discussing with you.

---

### Official Review · Reviewer_zHuG · 2022-07-11

**Rating:** 7
**Confidence:** 4
**Soundness:** 3 good
**Presentation:** 2 fair
**Contribution:** 3 good

**Summary:**

This paper studies an important and interesting topic in adversarial training, i.e., robust fairness issue. The robust fairness issue indicates that the adversarially trained model tends to perform well in some classes and poorly in other classes, creating a disparity in the robust accuracies for different classes. This paper focuses on the influence of perturbation radii on the robust fairness problem and finds that there is a tradeoff between average robust accuracy and robust fairness. To mitigate this tradeoff, the authors present a method to correct the lack of fairness in adversarial training.

**Questions:**

Does the phenomenon in Table 2 still exist for other datasets?

**Strengths And Weaknesses:**

Strengths:
1. This paper empirically finds a new phenomenon that AT with a larger perturbation radius will lead to a larger variance of class-wise robust accuracy while average robust accuracy is improved. Concretely, the authors use Madry and TRADES with different neural network architectures and attack algorithms to conduct extensive experiments to prove the generality of this new phenomenon. Experimental results consistently support their finding.
2. The authors provide a theoretical explanation for their experimental results through a linear model. Then, this paper also provides evidence for the existence of a tradeoff between robust accuracy and robust fairness.
3. The authors present a method to correct the lack of fairness of the AT. Simultaneously, the authors empirically and theoretically show the effectiveness of the proposed method to obtain better robust fairness without significantly altering the robust accuracy of the model.

Weaknesses:
1. The paper is well-written overall, but it would benefit from a background section (before Section 3) that introduce some of the terminology and definitions for readers who are not familiar with the adversarial training literature.
2. There is too little space devoted to explaining experimental results, it is expected to proivide a more detailed discussion and analysis.

---

> ### Author Response · Authors · 2022-08-02
> **Response to Reviewer zHuG**
>
> Dear Reviewer zHuG,
>
> We thank the reviewer for the encouraging comments and meaningful feedback. We provide answers to the raised questions/cons below
>
> **(Q6): Definitions and experimental analysis.**
>
> We have given some necessary notations and definitions in section 3 before empirical exploration. Additionally, we add more discussion and analysis about experimental results in Appendix.
>
> **(Q7): Does the phenomenon in Table 2 still exist for other datasets?**
>
> Yes, the same phenomenon in Table 2 still exist for CIFAR-100. The table below show the
>  robust accuracy (\%), average robust accuracy (\%) and variance of class-wise robust accuracy ($10^{-2}$) of some classes for Madry using Wide ResNet-32-10 on CIFAR-100. The perturbation radii for AT are chosen from $\epsilon_{train}=\{4/255,8/255,12/255,16/255\}$.  The adversarial testing examples are generated by FGSM with testing perturbation radius $\epsilon_{test}=16/255$. We use var and rob.acc to denote the variance of class-wise robust accuracy, and average robust accuracy, respectively. Moreover, diff means the difference between testing results where $\epsilon_{train}$ is $16/255$ and $4/255$, respectively.
>
> | $\epsilon_{train}$ |beaver | shark | trout | skyscraper | var  | rob.acc. |
> |  :----:  | :----:  | :----:  | :----:  | :----:  | :----:  | :----:  |
> | 4/255  |   38.5   | 30.2   | 3.7  |   15.1     |1.65   | 15.87 |
> |8/255   |   45.3   |  35.0  |  2.3  |  13.6   |  2.22   |  19.94 |
> | 12/255|   46.4  |  39.1  |   2.0  |  11.4    |  2.32   |  20.52 |
> | 16/255 |  48.6 |     44.7 |   1.3   |  7.7     |  2.52  |  21.68  |
> | diff |    10.1  |   14.5    |   -2.4  |  -7.4   |  0.87   |  5.81  |

---

> ### Author Response · Authors · 2022-08-08
> **Discussion**
>
> Dear reviewer zHuG,
>
> we will really appreciate it if the reviewer can go over our detailed response. Please feel free to ask us any questions you may still have and we will be more than happy to answer them.
>
> Thank you again for reviewing our paper and we look forward to discussing with you.

---

### Official Review · Reviewer_F5Kr · 2022-07-14

**Rating:** 3
**Confidence:** 3
**Soundness:** 3 good
**Presentation:** 1 poor
**Contribution:** 2 fair

**Summary:**

The authors investigate the robust fairness phenomenon in adversarial training and study the tradeoff between average robustness and robust fairness.  They find that with increasing perturbation radii the average robust accuracy improves over classes while the class-wise robust accuracy suffers from increased variability.  The authors propose a Fairly Adversarial Training (FAT) approach to mitigate the tradeoff between robustness and fairness in adversarial training.

**Questions:**

What is the significance of the results and how can this work impact other areas?

**Limitations:**

see above

**Strengths And Weaknesses:**

The general problem is clear and seems an important problem to address.

The authors should provide more motivation for why the problem is important to explore.  An example would be helpful here.

Section 3 is difficult to read and the definitions in section 4 should come before this section.  The experiments used to generate the results in this section also need much more information in regards to the data, application, models and training/evaluation strategy for this to be self-contained and fully convincing.

More detail on explaining FAT and how it is used in section 5 is needed.  This section is not clear.

The experiment section should be expanded in the paper or appendix to examine in more detail the significance of the results.  This is not currently clear.

It is not clear how large the impact this work will have in its current state.

---

> ### Author Response · Authors · 2022-08-02
> **More Explanations On Fairness and Experiments**
>
> Dear Reviewer F5Kr,
>
> Thank you for providing your helpful and constructive feedback. Sorry for the potential confusion in our paper and we would like to clarify them in our response. Following your comments, we have fixed the issues in revision (see text highlighted in red).
>
> **(Q1): Why is the problem studied in this paper important? give an example to explain.**
>
> Robust fairness issue in adversarial training recently draws much attention from the community of machine learning (see inference
>  in line 24). Fairness is very important. For example, an autonomous driving system may achieve a satisfying average robust accuracy for recognizing objects in the road. Nevertheless, this system is robust to the classes of inanimate objects (high robust accuracy) but vulnerable to some important classes such as “human” (low robust accuracy), which is the snake in the grass for drivers and pedestrians in the road. Additionally, in many sensitive fields related to social ethics, it is crucial to ensure that the models have no discriminatory behaviors toward certain classes (groups or populations). It is therefore imperative to conduct in-depth study on the robust fairness issue. We have added these examples to Section 1 in new revision (line 28 - line 35 ).
>
> **(Q2): Definitions should come before section 3 and provide more information about data, models and training/evaluation strategy.**
>
> We are so sorry to confuse you for reading. In new revision, we have provided necessary notations and definitions in section 3 before empirical exploration. The experiments in section 4 (in new revision, same below) are used to explore the effect of different perturbation radii in AT on robust fairness (the variance of class-wise robust accuracy) through popular neural networks, adversarial training strategies and attack algorithms. We use Madry(PGD-10) and TRADES($\lambda =6$) with various perturbation radii $\epsilon_{train}=\{4/255,8/255,12/255,16/255\}$ to train ResNet18, ResNet50 and Wide ResNet-32-10 (WRN-32-10) on the CIFAR-10 and CIFAR-100 datasets. The evaluation metrics are the average robust accuracy and the variance of class-wise robust accuracy. The adversarial testing examples are generated by FGSM, PGD-20 and CW with different testing perturbation radii $\epsilon_{test}=\{16/255,20/255\}$, respectively. The experimental settings including optimizer, epoch, learning rate and so on in section 4 are same as section 7.1. more detailed information about Mary, TRADES, attack algorithms and datasets can be found in related reference (line 97 - line 102).
>
> **(Q3): Explain FAT and how it is used in section 6.**
>
> Sorry for the confusion. Motivated by the theoretical results in section 5, we find that controlling the variance of class-wise robust accuracy is equal to control the variance of class-wise adversarial risk. Moreover, Theorem 6.1 suggests that adversarial risk can be bounded by empirical adversarial risk and the variance of class-wise adversarial risk. Therefore, we propose a variance-based regularized method FAT to mitigate the fairness issue. Then, we explain the training process as follows:
>
> (1) Using PGD-10 to generate a batch of adversarial examples, the batch size is 512 for Madry and 256 for TRADES, respectively.
>
> (2) $n_i$ is the number of adversarial examples that belong to class $i$ in the batch. Compute the losses per adversarial sample, and then we have the empirical class-wise average loss
>  $\hat{R}_{adv}(f,i) = \frac{1}{n_i} \sum^{n_i} _{j=1} \tilde{\ell} (\theta, x_j, i) $;
>
> (3) Compute VCAR and minimize the object  $\hat{R}_{adv}(f) + \lambda  \widehat{\operatorname{VCAR}}(f)$.
>
> **(Q4) More experiments in the paper or appendix.**
>
> We represent the experimental results of Madry and FAT on CIFAR-10 in section 7. the experimental results of Madry and FAT on CIFAR-100 can be found In Appendix, Simultaneously, we also provide the experimental results of TRADES and FAT through three neural networks ResNet-18, ResNet-50 and Wide ResNet-32-10 on CIFAR-10. All experimental results consistently support our new findings and theoretical results, namely there exists a tradeoff between robustness and fairness, and FAT can effectively mitigate the fairness issues.
>
> **(Q5) What is the significance of the results and how can this work impact other areas?**
>
> Our experimental and theoretical results suggest that only pursuing a higher robust accuracy will lead to severer robust fairness issues by sacrificing model’s robust accuracy on relatively vulnerable classes, which will lay serious hidden dangers for some applications. Some examples can be seen in answer of Q1.

---

> ### Author Response · Authors · 2022-08-06
> **Discussion**
>
> Dear reviewer  F5Kr,
>
> we will really appreciate it if the reviewer can go over our detailed response and revisions. Please feel free to ask us any questions you may still have and we will be more than happy to answer them.
>
> Thank you again for reviewing our paper and we look forward to discussing with you.

---

> ### Author Response · Authors · 2022-08-08
> **More Discussion**
>
> Dear Reviewer F5Kr,
>
> Thank you for reviewing our paper. Since your rating is low, we hope to have more discussion with you. If you have any other questions, please let us know.

---

### Meta-Review · Area_Chair_SvwS · 2022-08-24

**Recommendation:** Accept
**Confidence:** Less certain

**Metareview:**

While the reviews are a bit divergent, it seems many of the concerns raised are addressed properly and some of them are confirmed with the reviewer. I believe the paper is well-written and the contribution is clear with supporting experiments and theory. Hence, I recommend the acceptance of this paper.

**Award:**

No

---

### Decision · Program_Chairs · 2022-09-14

Accept